# A spatial evaluation of high-resolution wind fields from empirical and dynamical modeling in hilly and mountainous terrain

Christoph Schlager[1], Gottfried Kirchengast[1,2], Juergen Fuchsberger[1], Alexander Kann[3], and Heimo Truhetz[1]

[1]Wegener Center for Climate and Global Change (WEGC), University of Graz, Graz, Austria.
[2]Institute for Geophysics, Astrophysics, and Meteorology/Institute of Physics, University of Graz, Graz, Austria.
[3]Department of Forecasting Models, Central Institute for Meteorology and Geodynamics (ZAMG), Vienna, Austria.

**Correspondence:** Christoph Schlager (christoph.schlager@uni-graz.at)

**Abstract.** Empirical high-resolution surface wind fields, automatically generated by a weather diagnostic application, the WegenerNet Wind Product Generator (WPG), were intercompared with wind field analysis data from the Integrated Nowcasting through Comprehensive Analysis (INCA) system and with regional climate model wind field data from the Consortium for Small Scale Modeling Model in Climate Mode (CCLM). The INCA analysis fields are available at a horizontal grid spacing of 1 km × 1 km, whereas the CCLM fields are from simulations at a 3 km × 3 km grid. The WPG, developed by Schlager et al. (2017, 2018), generates diagnostic fields at a high resolution grid of 100 m × 100 m, using observations from two dense meteorological station networks: The WegenerNet Feldbach Region (FBR), located in a region predominated by a hilly terrain and its alpine sister network, the WegenerNet Johnsbachtal (JBT), located in a mountainous region.

The wind fields of these different empirical/dynamical modeling approaches were intercompared for thermally induced and strong wind events, using hourly temporal resolutions as supplied by the WPG, with the focus on evaluating spatial differences and displacements between the different datasets. For this comparison, a novel neighborhood-based spatial wind verification methodology based on fractions skill scores (FSS) is used to estimate the modeling performances. All comparisons show an increasing FSS with increasing neighborhood size. In general, the spatial verification indicates a better statistical agreement for the hilly WegenerNet FBR than for the mountainous WegenerNet JBT. The results for the WegenerNet FBR show a better agreement between INCA and WegenerNet than between CCLM and WegenerNet wind fields, especially for large scales (neighborhoods). In particular, CCLM clearly underperforms in case of thermally induced wind events. For the JBT region, all spatial comparisons indicate little overlap at small neighborhood sizes and in general large biases of wind vectors occur between the regional climate model (CCLM) and analysis (INCA) fields and the diagnostic (WegenerNet) reference dataset.

Furthermore, gridpoint-based error measures were calculated for the same evaluation cases. The statistical agreement, estimated for the vector-mean wind speed and wind directions show again a better agreement for the WegenerNet FBR than for the WegenerNet JBT region. A combined examination of all spatial and gridpoint-based error measures shows that CCLM with its limited horizontal resolution of 3 km × 3 km and hence, a too smoothed orography, is not able to represent small-scale wind patterns. The results for the JBT region indicate significant biases in the INCA analysis fields, especially for strong wind speed events. Regarding the WegenerNet diagnostic wind fields, the statistics show acceptable performance in the FBR and somewhat overestimated wind speeds for strong wind speed events in the Enns valley of the JBT region.

## 1 Introduction

Surface wind is often considered as one of the most difficult meteorological variables to model, particularly over areas of complex terrain like the Alps (Whiteman, 2000; Sfetsos, 2002; Abdel-Aal et al., 2009; Gómez-Navarro et al., 2015). Therefore,
realistic high-resolution wind fields cannot be generated with coarse-resolution models or by a simple interpolation of wind station data onto regular grids. Innovation in computer sciences, new methods in weather analysis or nowcasting models, advanced software architectures used in regional climate models (RCMs) and the growing power of computers meanwhile led to highly-resolved outputs from such models at the 1-km scale (Awan et al., 2011; Suklitsch et al., 2011; Prein et al., 2013b, 2015; Leutwyler et al., 2016; Kendon et al., 2017).

These models, however, contain various limitations and sources of uncertainties. In case of weather analysis fields, which is a mixed empirical and dynamical modeling from data assimilation, they result from too little meteorological station and remote sensing data, and in case of regional climate models (RCMs) they include deviations in the driving data set, physical and numerical approximations, as well as parameterizations of processes at the sub-grid scale (Gómez-Navarro et al., 2015).

To evaluate and improve these analysis and models, meteorological observations and especially gridded empirical datasets
at high spatial and temporal resolutions are needed. The model outputs on their side generally represent the involved processes as areal averages rather than on a point-scale (Osborn and Hulme, 1998; Prein et al., 2015). Therefore, gridded meteorological evaluation datasets, with each (aggregated) grid value being a best-estimate average of the grid cell observations, are the most appropriate evaluation datasets (Haylock et al., 2008; Haiden et al., 2011; Hiebl and Frei, 2016).

To investigate weather and climate on a local 1-km scale as well as evaluating RCMs, the Wegener Center at the University
of Graz operates two high-resolution meteorological station networks (Fig. 1): the very high-density WegenerNet Feldbach Region (FBR) in southeastern Styria, Austria (Fig. 1b) and the high-density WegenerNet Johnsbachtal (JBT) in northern Styria, Austria (Fig. 1c); details introduced in section 2 below.

For both networks, diagnostic windfields at a high resolution grid of 100 m $\times$ 100 m are generated by a weather diagnostic application, the WegenerNet Wind Product Generator (WPG). Schlager et al. (2017) introduced the WPG and its performance
evaluation for the WegenerNet FBR, which was then advanced by Schlager et al. (2018) to the WegenerNet JBT region and a longer-term evaluation in both the FBR and JBT regions. Jointly these studies established the level of quality of the empirical WPG wind fields. In this study, we now make use of these empirical high-resolution wind fields as reference data in order to intercompare them with empirical-dynamical wind field analysis data and with dynamical regional climate model data.

In this study, we intercompare the empirical WegenerNet wind fields (Schlager et al., 2017, 2018) with empirical-dynamical
wind field analysis data from the Integrated Nowcasting through Comprehensive Analyses (INCA) (Haiden et al., 2011) system and with regional climate model data from the Consortium for Small Scale Modeling Model in Climate Mode (CCLM) (Böhm et al., 2006; Rockel et al., 2008). The intercomparisons aim at getting useful and robust information about performance limits for these empirical and dynamical modeling approaches for regions with very different topographic characteristics and

weather situations. Furthermore, we co-analyze the impact of different horizontal resolutions, which inevitably will always be a challenge for the wide diversity of data products typically available.

Besides traditional gridpoint based verification methods, we use a novel wind verification methodology, recently developed by Skok and Hladnik (2018). This neighborhood-based spatial verification method avoids the "double penalty" problem and can distinguish forecasts depending on the spatial displacement of wind patterns (Skok and Hladnik, 2018). A 'double penalty' problem arises when using traditional statistical methods for datasets which contain an offset between the modeled and the reference data. In that case, the modeled data are penalized twice: first, for simulating an event where it did not occur and second, for failing to simulate an event where it did actually occur (Roberts, 2008; Prein et al., 2013a; Skok and Hladnik, 2018). So our primary motivation of this study is indeed to explore and provide improved insight, by careful intercomparisons the relative performance strength and weakness of empirical an dynamical wind field modeling at high-spatial resolution over complex terrain where actual wind station observations will generally be available at sparse station density.

The paper is structured as follows. Section 2 provides a description of the study areas and basic information about the model data. Section 3 presents defined evaluation cases and the methodology for the automatic selection of typical wind events followed by a description of the methods used to evaluate model results. In the following section 4 results are presented and discussed in detail. Finally, in Section 5 we summarize our results and draw our conclusions.

## 2   Study Areas and Model Data

### 2.1   Study Areas

The first study area, the WegenerNet FBR (indicated by the lower white-filled rectangle in Fig. 1a, enlarged in b) lies in the Alpine foreland of southeastern Styria, Austria, centered near the town of Feldbach (46.93°N, 15.90°E). It covers a dense grid of 154 meteorological stations within an area of about 22 km × 16 km, in a hilly terrain, characterized by small differences in altitude (Kirchengast et al., 2014). The typical difference in altitude between the valleys and the crests is about 100 m and the highest peak is the Gleichenberg Kogel, with an elevation of 598 m.

This region, with a more Alpine climate at the valley floors and more Mediterranean climate along hillsides is quite sensitive to climate change (Wakonigg, 1978; Kabas et al., 2011; Hohmann et al., 2018). Furthermore, it exhibits rich weather variability, especially through strong convective activity and severe weather in summer (Kirchengast et al., 2014; Kann et al., 2015a; O et al., 2017, 2018; Schroeer and Kirchengast, 2018). The wind fields in this study area are characterized by thermally induced local flows and influenced by thermally-driven regional wind systems with weak wind speeds, caused by a dynamical process called Alpine pumping (Lugauer and Winkler, 2005). Furthermore, nocturnal drainage winds, which are leading to cold air pockets, are relevant for this region, which is dominated by agriculture. Especially in fall and winter, the nocturnal cold air production is amplified by temperature inversions in relation to high-pressure weather conditions. In the WegenerNet FBR, hillside locations are thermally preferred to valley locations at night. Results related to the WPG-diagnosed empirical wind fields in the WegenerNet FBR can be found in Schlager et al. (2017, 2018).

The second study area, the WegenerNet JBT (indicated by the upper white-filled rectangle in Fig. 1a, enlarged in c), is named after the Johnsbachtal river basin (location of village Johnsbach 47.54°N, 14.58°E) and situated in the eastern Alpine region, in the *Ennstaler Alps* and the *Gesäuse National Park*, in the northern Styria, Austria. The terrain of this mountainous region is characterized by large differences in elevation. The Hochtor, with an elevation of 2369 m, is the highest summit, and the valleys are roughly at a height from 600 m to 800 m (Strasser et al., 2013; Schlager et al., 2018). This region spans an area of about 16 km ×17 km and comprises 11 irregularly distributed meteorological stations including two summit stations at altitudes of 2.191 m and 1.969 m (Schlager et al., 2018).

The climate is Alpine with mean annual temperatures of around 8 °C to 0 °C and an annual precipitation of about 1.500 mm to 1.800 mm from the valley to the summit regions (Wakonigg, 1978; Prettenthaler et al., 2010). Typical for this region are thermally induced local flows and westerly-flow synoptic weather conditions. Details related to first studies and their results as well as to the cooperation and partnerships can be found in Strasser et al. (2013), and in most up-to-date form in Schlager et al. (2018). Recently, Schlager et al. (2018) computed and evaluated WPG-generated empirical wind fields in this region.

## 2.2 WegenerNet data

The data acquired from the two WegenerNet regions FBR and JBT are automatically quality controlled and processed by the WegenerNet Processing System (WPS), consisting of four subsystems (Kirchengast et al., 2014): The Command Receive Archiving System transfers raw measurement data via wireless transmission to the WegenerNet database in Graz, the Quality Control System checks the data quality, the Data Product Generator (DPG) generates regular station time series and gridded fields of weather and climate products, and the Visualization and Information System offers the data to users via the Wegener-Net data portal (www.wegenernet.org).

Besides weather and climate time series, the DPG generates, based on a spatial interpolation of the station observations, gridded fields of the variables temperature, precipitation and relative humidity for the WegenerNet FBR. These gridded products of the WegenerNet FBR are available to users in near-real time with a latency of about 1-2 hours. Kirchengast et al. (2014) and Kabas (2012) provide detailed information about the subsystems of the WPG.

The DPG furthermore includes a newly developed wind field application, the Wind Product Generator (WPG), as briefly introduced in Sect.1. The WPG provides high-resolution wind fields for the WegenerNet FBR as well as for the WegenerNet JBT. The WPG uses the freely available empirical California Meteorological Model (CALMET) as core tool and generates, based on meteorological observations, terrain elevations and information about land use, mean wind fields at 10 m and 50 m height levels with a spatial resolution of 100 m × 100 m and a temporal resolution of 30 minutes, again with a maximum latency of about 1-2 hours. In order to keep the meteorological input data of the WPG independent from the data pertaining to the other operational station networks, observations from the ZAMG stations (violet stars in Fig. 1a and Fig. 1b) and other external stations are not used as WPG input. For the WegenerNet FBR, the gridded wind fields are available starting in 2007 and for the WegenerNet JBT starting in 2012. The wind fields at 10 m height level are used for the model intercomparisons.

The CALMET model is a diagnostic model that omits time-consuming integrations of nonlinear equations, such as the governing equations of dynamical models (Truhetz, 2010; Seaman, 2000; Ratto et al., 1994). It is hence not capable of simulation

of dynamic processes such as flow splitting and grid-resolved turbulence, or to deliver prognostic information. Specific parameterizations allow the model to empirically take into account conditions such as kinematic effects of terrain, slope flows, and terrain-blocking effects (Scire et al., 1998; Cox et al., 2005; Seaman, 2000). We enhanced the model by implementing methods developed by Bellasio et al. (2005) to as well take into account topographic shading through relief, topographic slope and aspect, and the sun position for the estimation of solar radiation. In addition, the modeling of temperature fields is now based on vertical temperature gradients, calculated from meteorological station observations located at different altitudes, and the influence of vegetation cover is taken into account. Details about these advanced algorithms can be found in Bellasio et al. (2005).

The quality of the generated wind fields depends above all on the quality and the spatial and temporal resolution of the meteorological observations and surface-related datasets, which are used as model input (Schlager et al., 2017, 2018; Morales et al., 2012; Cox et al., 2005; Gross, 1996). A detailed description of the WPG application and the statistical results for the WegenerNet FBR can be found in Schlager et al. (2017). More information regarding statistical results related to the WegenerNet JBT as well as information regarding evaluation results from five-year climate data of the WegenerNet JBT in comparison to nine-year climate data from the WegenerNet FBR can be found in Schlager et al. (2018).

## 2.3 INCA data

The INCA system has been developed at the Central Institute for Meteorology and Geodynamics (ZAMG) in Vienna, Austria, to provide realistic analyses and nowcasts of quantities of several meteorological variables for the highly mountainous and the overall complex terrain of Austria. In case of the variable wind, the system operationally generates spatially distributed analysis wind fields in 3D and for 10-m height above ground with a horizontal grid spacing of 1 km $\times$ 1 km and a temporal resolution of 1 hour.

The basic idea of the INCA wind module is to statistically correct a numerical weather prediction (NWP) model first guess (i.e., in operational mode the latest available NWP model run) with latest observational data which are not entering the NWP data assimilation. Thus, the skill of the INCA analysis depends on the station density, their representativeness and the spatial distribution of station observations, as well as on the skill of the NWP model providing the first guess. The impact of the NWP model on the analysis skill is further discussed in Kann et al. (2015b).

In this study, NWP model outputs used as first guess for INCA were generated by the the spectral ARPEGE-ALADIN (ALARO) model in revised version (Wang et al., 2006). ALARO has a horizontal grid spacing of 4.8 km $\times$ 4.8 km (600 x 540 grid points) and includes 60 vertical layers up to the 2 hPa level (about 43 km altitude), covering Central Europe, Eastern France and the Northern part of the Mediterranean Sea. It is run with a temporal resolution of 180 s using a hydrostatic semi-implicit semi-Lagrange dynamical solver (Bubnová et al., 1995) and the ALARO-0 physics package, which includes the 3MT microphysics-convection scheme (Gerard and Geleyn, 2005), the ISBA force restore 2L soil scheme (Noilhan and Planton, 1989), and the ACRANEB radiation scheme (Ritter and Geleyn, 1992). Soil temperature and moisture are initialized by a 6h-cycle optimal interpolation data analysis taking into account the latest ALARO forecast as first guess and 2 m relative humidity

and temperature observations from SYNOP and national stations. The 2 m values are transferred to soil variables via empirical relations (Giard and Bazile, 2000). To reduce initial spin-up a digital filter initialization is applied.

The model gets its lateral boundary and atmospheric initial conditions from the high-resolution deterministic operational global integrated forecast system (IFS) of the European Centre for Medium-range Weather Forecasts (ECMWF) model in lagged mode (i.e., ALARO 00 UTC is linked to IFS 18 UTC of the day before, ALARO 06 UTC to IFS 00 UTC, etc.). This is due to the rather late availability of the IFS data. Coupling is achieved by one-way nesting via Davies relaxation (Davies, 1976). Sea surface temperature is interpolated from the deterministic IFS model to the ALARO grid. More details about ALARO development and configurations can be found in Termonia et al. (2018).

The INCA wind fields have already been evaluated for a moderately hilly region in the north of Austria (47.78°N–49°N, 13.8°E–17°E), where the wind analyses show significantly higher errors compared to the statistical results from other meteorological variables (Haiden et al., 2011). These higher errors mainly root in the limited representativeness of station data, as well as on the low station density, which can be only partly compensated by INCA's analysis algorithms (Haiden et al., 2011).

In the WegenerNet FBR area, INCA assimilates observations from the ZAMG Feldbach and Bad Gleichenberg stations (violet stars in Fig. 1b) to the NWP's first guess, and in the WegenerNet JBT region, observations from the ZAMG Admont station are used. However, data from WegenerNet FBR and JBT stations are not used in INCA data assimilation and hence the WPG fields can be used for independent evaluation (Haiden et al., 2011).

The coordinate system of the INCA datasets is transformed into WGS84 / UTM zone 33N coordinates. Furthermore, we resampled the wind fields at 10 m height levels from the INCA gridding onto the WegenerNet FBR and WegenerNet JBT grids, using a bilinear interpolation method. Based on extensive sensitivity tests regarding different interpolation methods, we concluded that the statistical results do not significantly depend on the interpolation method.

## 2.4 CCLM data

Regarding the CCLM (Rockel et al., 2008), we use available wind fields generated with the model version 5.0. These wind fields were generated during the course of a previous study and are available for the period 2008 to 2010. The data have a comparatively coarse horizontal resolution of 3 km × 3 km on an hourly basis that is nevertheless the highest resolution proper available to this study. This limited resolution leads to a smoothed orography, which may result in different wind patterns with errors in wind speed or direction. Furthermore, the winds may be displaced by an incorrect position of the topographic slopes (Skok and Hladnik, 2018; Prein et al., 2013b). The CCLM fields are provided for 40 vertical levels. The first level is simulated for 10 m above ground and the last level corresponds to the 100 hPa level (about 16 km altitude), whereby the vertical resolution is higher in the boundary layer and decreases towards to the top level.

CCLM is a non-hydrostatic model with a Runge-Kutta dynamical core, which makes use of a 3rd order scheme with diffusion damping to discretize the advection term in the compressible Euler equations (Wicker and Skamarock, 2002). In order to avoid numerical instability, the model's orography is additionally smoothed via a 10th-order Raymond (1988) filter. The vertical coordinate system is a terrain-following, time-invariant Gal-Chen pressure-based sigma coordinate (Gal-Chen and Somerville, 1975) Deep and shallow convection are parametrized following Tiedtke (1989) and turbulence is parameterized based on Mellor

and Yamada (Raschendorfer, 2001). Vertical mixing comes from a prognostic formulation of the turbulent kinetic energy (TKE) with a 2.5 closure that accounts for grid- and subgrid-scale water and ice clouds. It uses a statistical cloud scheme for cloud cover and cloud water content (so-called Gaussian closure scheme). Horizontal diffusion follows the Smagorinsky approach.

Land cover data for CCLM are based on the Global Land Cover 2000 project (EEA, 2016) from SPOT4 satellite products (Bartalev et al., 2003). In the model setup used (3 km resolution), deep convection is resolved explicitly, which means that parameterization for deep convection was switched off. Shallow convection is still parameterized. In climate research, such simulations are referred to as convection-permitting climate simulations (CPCSs) (Prein et al., 2013a). To minimize decoupling effects from model-internal variability, that usually occur in large model domains and if nudging techniques are not used (Kida et al., 1991), CCLM is operated in a small domain encompassing the Greater Alpine Region and it is also driven by ECMWF's IFS. The data assimilation system IFS includes a wide range of observations and is assumed to provide perfect boundary conditions with a horizontal grid spacing of about 25 km at mid latitudes, and on 91 vertical levels (Bechtold et al., 2008). Every 6 h (00, 06, 12, 18 UTC) of the IFS data is an analysis field from the assimilation system and every alternate 6 h (03, 09, 15, 21 UTC) is a short-range forecast field. This procedure has already been used by Suklitsch et al. (2011) and keeps the modeled synoptic patterns in agreement with the observed ones.

In course of the data preparation for the study, CCLM data at 10 m height level were also transformed into WGS84/UTM zone 33N coordinates, resampled and mapped onto the high-resolution WPG grid, and checked for sensitivity with respect to the interpolation method which was as well found weak (see Section 2.3 above).

## 3 Evaluation events and methods

### 3.1 Events for wind field evaluation

The WegenerNet, INCA and CCLM wind fields are intercompared for two representative types of wind events: thermally induced wind events and strong wind events. For this purpose, we defined eight evaluation cases, four for each of the two study areas (Table 1). For the cases shown in Table 1 we use the WegenerNet data as reference, except for evaluating the CCLM wind fields for the WegenerNet JBT. The reason for this is, that the CCLM data used in this study are available from 01/2008-12/2009, but the WegenerNet JBT data are available only as of 01/2012, since this latter network was sufficiently completed for long term monitoring only since 2012 (Table 1, cases CCLMvsINCA_therm_JBT and CCLMvsINCA_strong_JBT).

In both study areas, autochthonous weather conditions mainly lead to thermally induced wind systems, meaning that the wind fields are controlled by small-scale temperature and pressure gradients. These small-scale gradients lead to characteristic interacting systems of air motion, like slope winds and mountain-valley winds, and create complex everyday flow patterns. The autochthounous days are characterized by small synoptic influences, cloudless or nearly cloudless skies, low relative humidity and increased radiation fluxes between the Earth surface and the atmosphere (Prettenthaler et al., 2010). Due to frequently occurring temperature inversions in relation to clear sky and high pressure weather conditions in winter, which often leads to a stable atmospheric stratification in the whole WegenerNet FBR and in the valley regions of the WegenerNet JBT, autochthonous days are only selected from spring, summer and fall (March to October).

The automatic selection of thermally induced and strong wind events is done based on thresholds, that we defined based on sound physical and careful sensitivity checks summarized in Table 2. For the estimation of autochthonous days, we compared the observed daytime mean values of wind speed ($v$) and relative humidity ($rh$) as well as the nighttime mean values of net radiation ($Q_n$) from selected stations with the respective thresholds. A further criterion for the selection of such days is a high daily global radiation, which indicates fair weather conditions. For this purpose, we compared the daily mean modeled global radiation ($Q_{g,m}$) for clear sky conditions with the observed daily mean net radiation ($Q_{n,o}$) for the WegenerNet FBR and with the observed daily mean global radiation ($Q_{g,o}$) for the WegenerNet JBT at defined station locations (Table 2, reference data). The reason for the comparison of $Q_{g,m}$ with $Q_{n,o}$ for the WegenerNet FBR is that this station network includes no global radiation sensors. Due to the almost linear relationship between the daytime $Q_{g,o}$ and $Q_{n,o}$ for clear sky conditions we find that the same selection method can robustly be applied to both study areas by defining different thresholds (Table 2, $Q_{g,m}$-$Q_{n(g),o}$). If all criteria are fulfilled for a given day, the data from the entire day are added to the thermally induced wind events dataset, leading to 24 hourly events (i.e., 24 hourly-mean wind speed values).

The modeling of the global radiation is done based on ESRI's ArcGIS Area Solar Radiation Tool. This tool is designed for local landscape scales and derives the incoming solar radiation based on a digital elevation model. Small differences of daily mean values between $Q_{g,m}$ and $Q_{n(g),o}$ indicate fair weather conditions and high global radiations during the day. If all criteria are fulfilled for a given day, the data from that day are included into the thermally induced wind events dataset.

The strong wind events, caused by synoptic weather conditions such as cyclones and frontal system at larger scale, are selected on an hourly basis from preselected days, by comparing hourly mean values from gridded reference datasets with defined minimum wind speeds. These preselected days were estimated by comparing the daily average wind speed from the gridded datasets with a defined minimum average wind speed (Table 2, $\overline{v}$ and $v$ for strong wind speed cases). If the hourly-mean value of this reference dataset is larger than the defined minimum wind speed, the data of this reference dataset and the corresponding model dataset are used as part of the hourly event data for evaluating strong wind events.

## 3.2  Statistical evaluation methods

In order to account for spatial differences and displacements between the model and the reference data and to analyze wind speed and direction in a combined way, we apply a novel wind verification methodology. This methodology extends the Fractions Skill Score (FSS), a spatial verification metric developed by Roberts (2008), which is classified as neighborhood-based approach and originally used for verifying precipitation. The FSS is based on the assumption that a model is useful when the model data and the corresponding reference data show a similar spatial frequency of precipitation events, which alliviates the requirement of the models to predict the events at exactly correct positions, which is an unduly strict assumption. Furthermore, this metric avoids the 'double penalty' problem and provides a scale dependent information about the level of model skill (Gilleland et al., 2010; Roberts, 2008).

In order to obtain information of how the skill of a model varies with the spatial scale, the FSS is calculated for different neighborhood sizes. A neighborhood size of $n$ defines a square consisting of $n \times n$ grid points, i.e., it denotes the side length of the square (e.g., for $n$=5 the square contains 25 grid points). These squares of defined neighborhood sizes are moved as sliding

windows over the datasets, centered successively at each grid point, whereby the area outside the domain is assumed to contain no windclass. In terms of FSS value, it is a 0-to-1 normalized metric, i.e., the lower limit of the FSS is 0 and the upper limit +1, with values approaching +1 representing a better degree of model performance.

The extended version of the FSS is named the Wind Fractions Skill Sore, denoted as WFSS which has been developed by Skok and Hladnik (2018). The score is calculated based on user defined wind classes. The definition of these classes is partly subjective and can significantly affect the WFSS. The wind vector field should be classified in such a way, that the definition reflects what a user wants to analyze. For example, a complex terrain leads to strong changes in wind directions, therefore it is reasonable to define smaller class intervals regarding the wind directions. For upper level winds the focus could be more on the magnitude of wind speed.

We defined eight wind direction classes with an interval size of $45°$ for a range of three wind speed categories, as shown in the windroses of Fig. 2. Wind speeds $< 0.5 \, \mathrm{m \, s^{-1}}$ were classified *calm*, independently of the wind direction. The small interval size of $45°$ was chosen to be able to capture the varying wind directions in the study areas, especially for thermally induced winds. Because of the generally much lower wind speeds in WegenerNet FBR we defined a smaller interval size of the wind speed categories for this region (Fig. 2a and b, lower panels) than for the WegenerNet JBT (Fig. 2c and d, lower panels).

Table 3 includes the equations for the calculation of the WFSS and the asymptotic WFSS (AWFSS). This AWFSS will always be reached for a neighborhood size $\geq 2N - 1$, where $N$ is the number of grid points of the largest domain size. At such a large neighborhood size, the estimated fractions within the domain are the same at all locations and further enlarging of this size will not affect the WFSS. A bias always leads to a AWFSS value < 1, which indicates systematic differences in the frequency of wind classes between the model and the reference wind classification.

The WFSS is calculated for each hourly field for the selected events. The event-averaged score values are calculated based on averaging these one-hour event WFSS values over all the hourly events within the analyzed multi-year period, for each evaluation case listed in Table 1.

As briefly explained above, the chosen thresholds of the classification and the number of classes are influencing the score. We found that especially the wind direction thresholds can have a strong impact on the score values. For example, a small change in the wind direction value from prevailing northwesterly winds, which are close to a threshold to distinguish between W and N, could dramatically change the WFSS value. Such a small error in the model data could indicate a poor modeling performance, whereas a human analyst would asses the forecast as reasonably good. To avoid this problem we calculated the hourly WFSS for every rotation between $0°$ and $45°$ with an interval size of $5°$ (nine trail classes), in addition to the original class definition. As next step, always the maximum values of the hourly score values at each neighborhood size are used for computing the final case-averaged score values. We applied this approach for in total 7597 selected events (Table 2, number of events) and estimated the case-averaged score value for each of the eight evaluation cases. A more detailed description of the wind fractions skill score metric can be found in (Skok and Hladnik, 2018).

Furthermore, we applied traditional gridpoint-based statistical performance measures such as bias, root-mean-square error and others to each evaluation case. All statistical performance metrics used in this study are summarized in Table 3.

## 4    Results

### 4.1    Evaluation for selected wind events

Figure 2a-d illustrates typical examples of modeled wind fields (upper rows of panels) and the corresponding windroses of relative frequency of wind directions divided by wind speed categories (lower rows of panels) from selected representative evaluation events. Each panel depicts modeled and the associated reference data for the WegenerNet FBR (Fig. 2a and b) and the WegenerNet JBT (Fig. 2c and d), for thermally induced (Fig. 2a and c) and strong wind events (Fig. 2b and d). Figure 2e shows the WFSS values of these examples, estimated as explained in Section 3.2 above.

The thermally induced wind event on the 29th of July 2009 from 16:00-17:00 UTC for the WegenerNet FBR (Fig. 2a) shows thermally driven regional flows caused by the Alpine pumping. This flow is called Antirandgebirgswind which arises usually in the afternoon as southerly wind with maximum wind speeds of about $2.5 \mathrm{~m~s}^{-1}$. The Antirandgebirgswind is a compensating flow between the bordering mountains of the eastern Alps, and the hilly countryside region of southeastern Styria (called Riedelland), which is comprising the WegenerNet FBR (Wakonigg, 1978). The INCA (upper-left panel of Fig. 2a) and the WegenerNet wind fields (upper-right panel of Fig. 2a) show a similar distribution with generally low wind speeds and prevailing southerly wind directions. The intercomparison of these INCA with WegenerNet data for this event shows the largest WFSS values for all neighborhood sizes, which indicates a good overlap of the wind classes (Fig. 2e, INCAvsWN_therm_FBR). Furthermore, it shows a nearly perfect asymptotic value of about 0.99. This large AWFSS indicates a very small bias, which is also reflected by the similar wind classification results (lower-left and lower-right panel of Fig. 2a).

The CCLM wind field shows similarly low wind speeds, compared to the WegenerNet wind field, but a shift in wind directions from the S sector mainly to the SE and partly to E and NE sectors between the CCLM and the WegenerNet data can be observed (lower-middle and lower-right panel of Fig. 2a). This shift is reflected by small WFSS values at all neighborhood sizes, especially below a scale of 10 km. The AWFSS shows the largest value of all CCLM intercomparisons, but is still low compared to INCA evaluation cases, which indicates a large bias (Fig. 2e, CCLMvsWN_therm_FBR). Evidently, this regional climate wind field modeling at 3-km gridsapcing is not adequately representative for the given challenging hilly terrain.

The strong wind speed event in the WegenerNet FBR on the 30th of October 2008 from 10:00-11:00 UTC led to southwesterly to southerly winds (Fig. 2b). The INCA model data and the WegenerNet reference data show maximum 1-hour vector-mean wind speeds of around $9-10 \mathrm{~m~s}^{-1}$ (upper-left and right panel of Fig. 2b). Regarding the wind directions, differences in the wind sectors can be observed (lower-left and right panel of Fig. 2b). The INCA data show wind directions mainly from the SW sector (lower-left panel) while in the WegenerNet data show wind directions from the S and SW sectors (lower-right panel). The WFSS for this case shows small values at small neighborhood sizes and increases with increasing neighborhood size (Fig. 2e, INCAvsWN_strong_FBR). These low WFSS values are mainly caused by the differences in wind direction classes, especially in the southern part of the domain and through some spatial displacements in wind speed classes. Despite low WFSS values at small neighborhood sizes caused by differences in wind sectors, the AWFSS shows a high asymptotic value (AWFSS>0.97). This high value is caused by the prevailing wind directions in the WegenerNet data, which are close to the threshold values to distinguish between S and the SW. In this case, the 5° azimuthal class rotation procedure hence avoids lower score values.

Regarding the CCLM data (lower-middle panel of Fig. 2b), the whole wind field shows wind speeds from about $6.5 \mathrm{~m~s}^{-1}$ to $7.5 \mathrm{~m~s}^{-1}$ and is therefore assigned to the wind class with wind speeds higher than $6 \mathrm{~m~s}^{-1}$. Whereas, for the WegenerNet wind fields, a large proportion is assigned to the class with wind speeds from $3 \mathrm{~m~s}^{-1}$ to $6 \mathrm{~m~s}^{-1}$ of this region (Fig. 2e, CCLMvsWN_strong_FBR) and indicates that the dynamically modeled CCLM wind speeds are systematically overestimated relative to the empirically diagnosed wind speeds.

This discrepancy leads to the smallest WFSS values at all neighborhood sizes for this region (Fig. 2e, CCLMvsWN_strong_FBR) and indicates that the dynamically modeled CCLM wind speeds are systematically underestimated relative to the empirically diagnosed wind speeds for this hourly event.

On the 1st of August 2012 the winds were thermally driven and the local pressure and temperature gradients were causing varying wind speeds and wind directions in the WegenerNet JBT. This is illustrated for the late afternoon INCA and WegenerNet wind fields in the upper left panels of Fig. 2c. The WFSS for the evaluation of the INCA wind field shows the second largest value at the 1 km neighborhood size, which indicates overlapping areas at this neighborhood size, equal to the horizontal resolution of the INCA analysis (Fig. 2e, INCAvsWN_therm_JBT). The large AWFSS value indicates again a small bias, which is also reflected by the similar wind classification shown in the windroses of the corresponding lower left panels of Fig. 2c. The high asymptotic value (AWFSS>0.9) indicates a small bias and that the WFSS is mostly influenced by the spatial displacement.

In a further example of a thermally induced wind event on the 31st of May 2008, we intercompare the CCLM with INCA wind fields (right panels of Fig. 2c). Especially in the CCLM, the smoothed terrain leads to uniform wind speeds and directions. Regarding the INCA wind fields, some variability in wind speed, with higher values in the summit regions and lower values at lower altitudes in the valleys of this region, can be observed. Furthermore, a valley wind in the Enns valley is simulated by INCA. Probably the analysis part of the INCA model with its higher-resolved DEM and assimilated ZAMG observations leads to a somewhat better representation of the wind field. Comparing wind directions, the largest part of the CCLM modeled flow is from the N and NE sectors, while the INCA system estimated wind directions mainly from the E sector and partly from the NE and SE sectors (bottom right panels of Fig. 2c). This simplistic pattern of wind directions in CCLM leads to low WFSS values for all neighborhood sizes, including the lowest asymptotic value of all examples, indicating a very poor representation of the wind field by the dynamical modeling of the CCLM in this challenging mountainous terrain.

The strong wind speed event for the WegenerNet JBT on the 7th of December 2013 is caused by northwesterly weather conditions. These synoptic scale flow conditions led to strong wind speeds with maximum 1-hour mean wind speeds of around $20 \mathrm{~m~s}^{-1}$ from 17:00-18:00 UTC. Both the INCA and the WegenerNet wind fields show wind directions mainly from the NW, with some proportions from the N and the W sectors, caused by a channeling of the air flow through the pronounced valleys of this study area. The INCA wind fields show much lower wind speeds in the valley regions compared to the WegenerNet wind fields, resulting from the observations of the ZAMG ADM station that flow into the INCA analysis but are considered far off the area and not used by the diagnostic modeling (upper left panels of Fig. 2d). As the neighborhood size increases, the WFSS also increases, but due to spatial displacements, the values are generally low (Fig. 2e, INCAvsWN_strong_JBT). The low AWFSS value is caused by the differences in wind speed categories (lower left panels of Fig. 2d).

For the 5th of November 2008 we intercompare the CCLM wind fields with INCA wind fields from 01:00-02:00 UTC, for a strong wind speed event (right panels of Fig. 2d). In this example, the influence of the smoothed terrain caused by the course horizontal resolution of the CCLM becomes obvious. This smoothed topography results in systematically lower wind speeds compared to the INCA wind fields. The WFSS shows similar results like for the previous INCAvsWN_strong_JBT evaluation, with small values at all neighborhood sizes (Fig. 2e, CCLMvsINCA_strong_JBT), indicating the clear limits of the CCLM dynamical modeling fields also for strong wind events.

## 4.2 Statistical evaluation results

The statistical event-averaged WFSS values from the large ensemble of events over multiple years are represented for each evaluation case in Fig. 3. Overall, it shows a monotonic increase with neighborhood size for all cases so that the AWFSS is the largest value, indication relatively the best performance at large scales.

For the WegenerNet FBR, the statistical WFSS values, calculated for the INCA wind fields compared to the WegenerNet wind fields, shows for both the thermally induced and strong wind events nearly the same behavior (Fig. 3a, INCAvsWN_therm_FBR and INCAvsWN_strong_FBR). The WFSS values for these cases indicate a reasonably good spatial matching at all neighborhood sizes. Furthermore, the AWFSS values are higher than 0.8, reflecting generally small INCA biases of wind classes.

The statistical WFSS estimated for the CCLMvsWN_therm_FBR case indicates that the CCLM clearly and systematically underperforms in case of thermally induced wind events for the WegenerNet FBR. Evidently, due to the coarse horizontal resolution of the wind fields, the CCLM wind fields appear fundamentally unable to capture the varying wind directions for such events in this region. For the CCLMvsWN_strong_FBR case, however the results indicate a similar spatial matching as for the INCAvsWN_therm_FBR and INCAvsWN_strong_FBR cases, just with a somewhat higher bias of wind class differences. This similar performance despite the coarser horizontal resolution of the CCLM model is explained through a weaker influence of the terrain on the wind fields under strong wind conditions in this region.

Because of the challenging terrain of the WegenerNet JBT, the statistical WFSS values are generally low for this region, signalling large biases (Fig. 3b). These biases are indicated by low asymptotic values, which tend to be between 0.61 and 0.64, except for the INCAvsWN_strong_JBT case, which shows an even lower value (AWFSS=0.39).

The spatial displacement and the biases for the INCAvsWN_therm_JBT case are mainly caused by the differences in wind directions for these thermally induced wind events. Especially at small neighborhood sizes at the 1-km scale, WFSS values indicate large spatial displacements.

The INCAvsWN_strong_JBT case shows the lowest values at all neighborhood sizes, but this time caused by the differences in the wind speed categories. These low values are caused by the INCA-analyzed wind speeds, which, in case of strong winds, are overestimated in the summit regions and underestimated in the valley regions. Slightly overestimated WegenerNet wind speeds in the Enns valley are somewhat reinforcing the difference between the INCA and the WegenerNet wind speeds. These differences in wind speed especially in the valley and the summit regions become obvious from Fig. 4c and 4d and are discussed in further detail below.

The intercomparison of the CCLM wind fields with INCA delivers nearly the same (low) WFSS values for both type of wind events. In case of thermally induced events (CCLMvsINCA_therm_JBT) the spatial displacements and biases are mainly caused due to differences in wind directions. For strong wind events (CCLMvsINCA_strong_JBT), the smoothed terrain caused by the coarse resolution of the CCLM leads to systematically underestimated wind speeds.

Table 4 summarizes, in addition to the AWFSS values, the results estimated with traditional statistical methods of the INCA analysis and CCLM dynamical fields. Due to the less challenging region of the WegenerNet FBR, all traditional statistical parameters show better performance for this region compared to the WegenerNet JBT. The absolute-value statistical metrics (bias $B$, standard deviation $SD_o$, root-mean-square-error $RMSE$) applied to the hourly vector-mean wind speeds, show higher values for the WegenerNet JBT, resulting from the generally higher wind speeds in addition to effects of the complex moun-
tainous terrain on the wind fields in this region. The $B$ values are slightly positive for the WegenerNet FBR and negative for the WegenerNet JBT. The substantially negative $B$ value for the INCAvsWN_strong_JBT case again reflects the underestimation of wind speed in the valleys, as explained above. Furthermore, the CCLMvsINCA_strong_JBT intercomparison also shows a negative bias, caused by the coarse resolution of the CCLM, which leads to lower wind speeds for strong wind events.

The $RMSE$ values range from 0.79 m s$^{-1}$ to 1.85 m s$^{-1}$ for the WegenerNet FBR and from 1.3 to 8.6 m s$^{-1}$ for the
WegenerNet JBT. The high value of 8.6 m s$^{-1}$ for the INCAvsWN_strong_JBT case is caused by the underestimation of wind speed in the valleys as well as the overestimation in the summit regions by the INCA model. The mean $R$ values show a better correlation for the WegenerNet FBR than for the WegenerNet JBT. The mean absolute error of wind direction ($MAE_{\mathrm{dir}}$) applied to hourly vector-mean wind directions also shows better performance (INCA and CCLM fields) for the WegenerNet FBR. Due to the varying wind directions caused by thermally induced circulations, the $MAE_{\mathrm{dir}}$ is higher for such events for
both study areas, with the highest value of 68° for the INCAvsWN_therm_JBT case.

Figure 4 shows the mean wind speed bias spatial distributions for all evaluation cases, for the WegenerNet FBR (Fig. 4a and b) and the WegenerNet JBT (Fig. 4c and d). The distribution for the INCAvsWN_therm_FBR case, the case for thermally induced wind events for the WegenerNet FBR, shows large areas with nearly no $B$ values (left panel of Figure 4a). The maximum $B$ value for this case can be observed in the area of the *Gleichenberger Kogel*, north of the ZAMG Bad Gleichenberg
station with a value of around 1.4 m s$^{-1}$. The evaluation of the CCLM for the same thermally induced events shows an overestimation of wind speeds for the whole study area, with $B$ values from 0.5 m s$^{-1}$ to 1 m s$^{-1}$ in the western part, and from 1 m s$^{-1}$ to about 1.4 m s$^{-1}$ in the eastern part of the study area (right panel of Figure 4a).

The overestimation of the wind speeds for the WegenerNet FBR can be explained by the too frequent flow-over patterns simulated for this region, which lead to a more dominant orographic speed-up effect. Due to the orographic smoothing, flow-over
patterns are generally more frequent than flow-around patterns, especially for the WegenerNet FBR with its small differences in altitude (Taylor et al., 1987).

The evaluation of the INCA model for strong wind speeds illustrates the strong influence of the terrain on this model. The results show a good agreement in the valleys of the study area, with partly small negative $B$ values (left panel of Figure 4b). The hilltop regions exhibit positive $B$ values, with maximum values of around 5 m s$^{-1}$ again in the area of the *Gleichenberger Kogel*.
Overall positive $B$ values of CCLM dynamical wind speed fields for strong wind events are seen in the right panel of Figure 4b,

showing the systematic overestimating by CCLM fields in this case. These large $B$-values are probably also due to the speed-up effect explained for the above case CCLMvsWN_therm_FBR.

For the WegenerNet JBT, the strong influence of the terrain on the INCA-analyzed wind speeds can be observed in all evaluation cases in this region. The evaluation of the INCA model for thermally induced wind events exhibit negative $B$ values in the valleys, whereby positive values are partly occurring in the summit regions (left panel of Figure 4c). At lower elevations, the intercomparison of the CCLM with the INCA model shows nearly no $B$ values for thermally induced events (right panel of Figure 4c). Furthermore, small negative $B$ values at the summit regions and some spots with positive values can be observed for this case. Due to these small bias values, similar results as these ones can be expected for a comparison of CCLM with WegenerNet data.

Similar bias distribution patterns as for the INCA evaluation for thermally induced wind events are present for strong wind events in the INCAvsWN_strong_JBT case, but this time with strong negative and positive $B$ values ranging from $-14.4\ \mathrm{m\ s^{-1}}$ to $4.9\ \mathrm{m\ s^{-1}}$ (left panel of Figure 4d). These strong negative $B$ values are again caused by the severely underestimated INCA wind speeds and the somewhat overestimated WegenerNet wind speeds in the valley regions of the WegenerNet JBT.

Opposite patterns can be seen for the intercomparison of the CCLM with the INCA model. This intercomparison exhibits small positive values in the valley regions and strong negative values in the summit regions (right panel of Figure 4d). Main reason for these strong negative $B$ values is the course resolution of the CCLM data and the resulting underestimation of wind speeds for strong wind events, as explained above. The negative $B$-values are likely caused by negative orographic speed-up effects, which are preferred in flow-around patterns and flow-splitting patterns that occur especially when the differences in the altitude of ridges of mountains are large (Hewer, 1998).

## 5 Conclusions

In this work we evaluated wind fields generated by two different modeling systems against empirically diagnosed wind fields from WegenerNet high-density network data: the INCA analysis system of the Austrian weather service ZAMG (Haiden et al., 2011) and the non-hydrostatic CCLM (Schättler et al., 2016). The INCA wind fields have a horizontal resolution of 1 km × 1 km, and in case of CCLM, 3 km × 3 km horizontal resolution was available, both on an hourly basis. The empirical high-resolution wind fields from the WegenerNet were generated by the WegenerNet Wind Product Generator (WPG), recently developed by Schlager et al. (2017, 2018).

The WPG-diagnosed gridded wind fields are available with a temporal resolution of 30 minutes and a spatial resolution of 100 m × 100 m and can therefore well serve as reference. The WegenerNet Feldbach Region (FBR) was used as study area, characterized by generally small differences in altitude in the hilly terrain of this region. The second study area was the WegenerNet Johnsbachtal (JBT) region, which is a mountainous region characterized through a very complex terrain.

The evaluation of the INCA and the CCLM wind fields was based on classifying the data separately into thermally induced and strong wind events. In case of the INCA evaluation, we could select wind events within the period 2008-2017 for the

WegenerNet FBR and within 2012-2017 for the WegenerNet JBT. For evaluating the CCLM data, events from the period 2008-2010 were selected for both study areas. Due to WegenerNet JBT wind fields being not yet available within 2008-2010, we intercompared the CCLM wind fields with INCA wind fields in this region.

Besides traditional performance measures such as bias, root-mean-square error, correlation coefficient, and mean absolute error of wind direction, we in particular applied a spatial wind verification methodology named the Wind Fractions Skill Score (WFSS) (Skok and Hladnik, 2018). This new score was used to detect spatial displacements of wind patterns and biases based on predefined wind speed and direction classes. The WFSS avoids the 'double penalty' problem and is able to distinguish between a 'near miss' and large displacements between modeled and reference wind fields. Furthermore, a spatial scale-dependent skill is determined by this score.

Due to the less challenging terrain of the Alpine foreland region, all statistical performance measures showed better INCA and CCLM performance for the WegenerNet FBR than for the challenging mountainous region WegenerNet JBT. The spatial verification of all evaluation cases indicates an increasing skill with increasing by larger scale (neighborhood size). For both study areas, the traditional statistical performance measures, applied to the wind speed, mostly show better performance of INCA and CCLM for thermally induced wind events than for strong wind events. On the other hand, the results related to wind direction indicate a better performance for strong wind events than for thermally induced events.

More specifically, the verification for the WegenerNet FBR shows that the INCA analysis wind fields are more skillful than the CCLM dynamical wind fields in this region. The INCA verification indicates a reasonably good performance for both thermally induced and strong wind events.

The CCLM clearly performs less well in case of thermally induced wind events for this region. The reason for this weak performance is the limited resolution of the wind field dataset from this model. Although the difference in the numerical resolution between INCA (1 km grid spacing) and CCLM (3 km grid spacing) is only a factor of 3, CCLM is not able to resolve small-scale wind patterns. This occurs for multiple reasons: 1) due to the $3^{rd}$-order advection scheme with its horizontal diffusion damping, the effective resolution in CCLM is several times coarser than the numeric grid spacing (Ogaja and Will, 2016); 2) the orography is smoothed as well, so that individual mountain ridge and valley structures are removed. For example, the mountain peak of the Hochtor with its 2396 m elevation in the center of the WegenerNet JBT region is lowered by about 500 m in the CCLM model.

Hence, with the resolution of 3 km $\times$ 3 km, the fields are not able to resolve the varying wind speeds and directions caused by thermally driven circulations. The wind speeds are overestimated by this model for both, thermally induced and strong wind events, and large differences in wind directions are found for thermally induced events.

For the WegenerNet JBT region, the verification shows generally large spatial displacements at all scales and strong biases in wind classes for all evaluation cases. In case of the INCA evaluation, large wind direction deviations for thermally induced wind events indicate that the analysis fields are not able to adequately capture the varying wind patterns such as slope and valley winds, which roots in the sparse station density that INCA can anchor to and the coarse horizontal resolution of the first guess provided by the ALARO model in this complex-terrain region. Furthermore, the statistics show an substantial underestimation of wind speeds in the valleys and overestimated wind speeds in parts of the summit regions for both type of wind events.

The intercomparison of the CCLM dynamical fields with the INCA analysis fields for thermally induced wind events are reflecting the disadvantage of smoothed terrain, which is caused by the limited effective resolution being several times coarser than the 3 km × 3 km grid spacing of the CCLM as already noted above. Improvements can be expected from latest developments in the numerical core of CCLM by Ogaja and Will (2016), which have enabled an improvement of the effective resolution by a factor of 2 via introducing a 4$^{th}$-order advection scheme that allows to circumvent the horizontal diffusion damping.

Based on these finding we suggest, underpinning the results of Haiden et al. (2011), that additional observed wind information in the summit and valley regions, especially in a complex terrain like the WegenerNet JBT, and a more comprehensive use of wind-constraining satellite data as well as a higher-resolution RCM could help to systematically improve the INCA-analyzed wind fields. At higher resolutions, the topographic shading through the terrain becomes increasingly important, especially for the simulation of thermally induced wind events. Such methods have not yet been implemented into the ALARO model, but may help to generate more realistic wind fields in the future.

Related to the CCLM dynamical modeling, a verification of CCLM-generated higher-resolution 1 km × 1 km wind fields and the application of the new 4th-order advection scheme from Ogaja and Will (2016) in a convection-permitting configuration would also be a promising issue for further investigations of how this may improve the modeling of wind patterns in a complex terrain.

Investigations regarding the WegenerNet JBT wind fields showed, that an additional wind-observing station in the Enns valley would improve the results for this region (Schlager et al., 2018). Such an additional station would avoid the overestimation of WegenerNet wind speeds in the Enns valley, especially for strong wind events. In the WegenerNet FBR region just recently (in May 2018) another wind station was added in the Raab valley (station Nr. 155 1b), which will further improve the WPG-derived fields in future. This adds further value to valuable reference for evaluation of important other data products such as the INCA operational analysis and dynamical climate model fields.

*Code availability.* The CALMET 6.5.0 model code is available from the website www.src.com/calpuff/. The INCA and the WPG code is not in the public domain and cannot be distributed. The source code for the CCLM is available on request via the website https://www.clm-community.eu. The code for the calculation of FSSwind score is available as part of the SpatialVx package (function calculate_FSSwind). SpatialVx is a R software package made by Eric Gilleland that enables the calculation of a large number of spatial verification scores (https://cran.r-project.org/package=SpatialVx).

*Data availability.* CORINE Land Cover data for the study area were taken from www.eea.europa.eu, digital elevation model data from www.gis.steiermark.at, and WegenerNet data from www.wegenernet.org. The WegenerNet data contain the WPG wind field output data as introduced in this study. The INCA data are available on request from the Central Institute for Meteorology and Geodynamics (klima@zamg.ac.at). CCLM data are available on request from the Wegener Center, University of Graz (heimo.truhetz@uni-graz.at).

*Author contributions.* C. Schlager collected the data, performed the analyses and modeling, created the figures, and wrote the first draft of the manuscript. G. Kirchengast provided guidance and advice on all aspects of the study and significantly contributed to the text. J. Fuchsberger provided guidance on technical aspects of the WegenerNet networks, and its data characteristics and contributed to the text. A. Kann provided INCA-related advice and contributed to the INCA part of the text and H. Truhetz provided information and advice on the CCLM setup and
5    characteristics and contributed in particular to the CCLM part of the text. All authors commented on the final version of the manuscript.

*Competing interests.* We declare that no competing interests are present.

*Acknowledgements.* The authors thank Gregor Skok (Dept. of Physics, Univ. of Ljubljana), for providing the R code to calculate the Wind Fractions Skill Score. Furthermore, the authors acknowledge the data providers at the Central Institute for Meteorology and Geodynamics (ZAMG) for the Integrated Nowcasting through Comprehensive Analysis (INCA) dataset. Andras Csaki (Wegener Center, Univ. of Graz) is
10    thanked for performing the CCLM modeling and extracting the wind field data. The authors are as well grateful to the Julich Supercomputing Centre (JSC) and the Vienna Scientific Cluster (VSC) for providing the necessary HPC resources. WegenerNet funding is provided by the Austrian Ministry for Science and Research, the University of Graz, the state of Styria (which also included European Union regional development funds), and the city of Graz; detailed information can be found online (www.wegcenter.at/wegenernet). The CCLM simulation was funded by the Austrian Science Fund (FWF) under project NHCM-2 (project number P24758-N29).

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

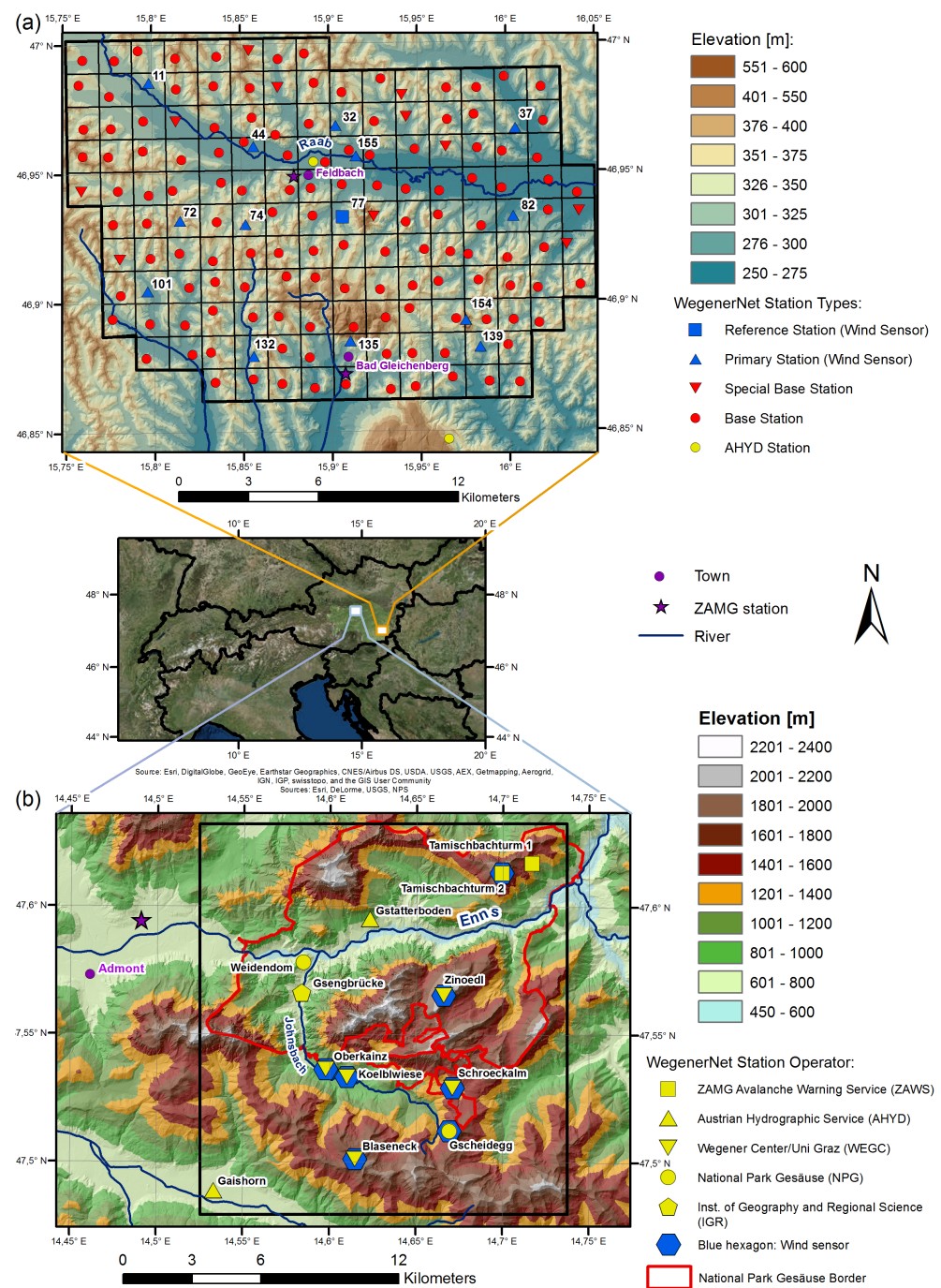

**Figure 1.** Location of the study areas in Austria (middle panel), the WegenerNet Feldbach Region (FBR) in the southeast of the state of Styria and the WegenerNet Johnsbachtal (JBT) region in the north of Styria (white-filled rectangles, enlarged in (a) and (b)). (a) The WegenerNet FBR with its 155 meteorological stations, with the legend explaining map characteristics and station types. (b) Map of the WegenerNet JBT region (black rectangle) including its meteorological stations, with the legend explaining map characteristics and station operators.

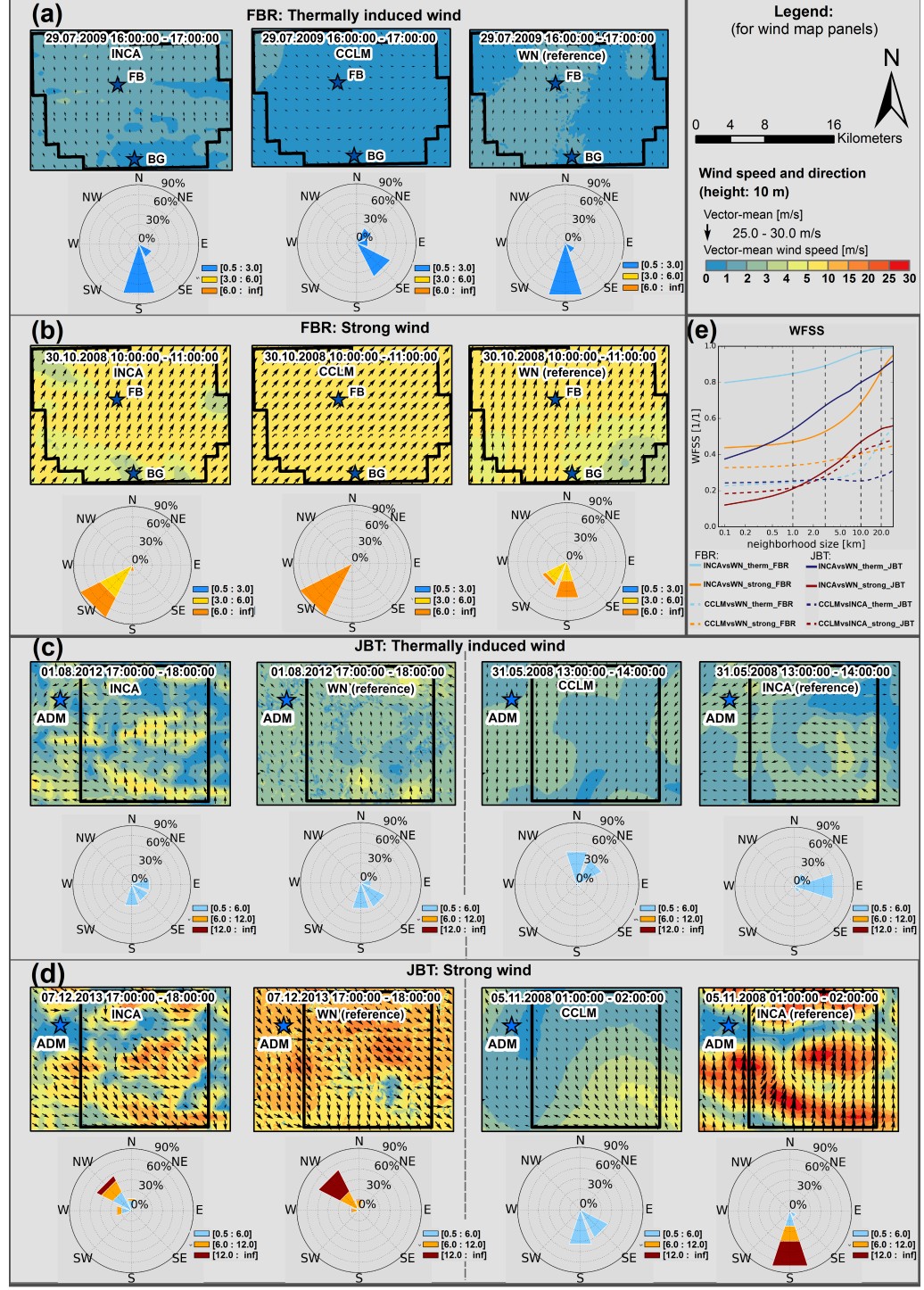

**Figure 2.** Wind Fractions Skill Score (WFSS) analysis for selected one-hour wind fields for the WegenertNet FBR (a, b) and the WegenerNet JBT region (c, d). (a-d) Modeled and reference wind fields (first row) and corresponding relative frequency of wind directions for a range of wind speed categories (second row), in each panel. (e) WFSS results for the modeled versus reference wind fields from (a)-(d). See Table 1 for more information on the evaluation cases.

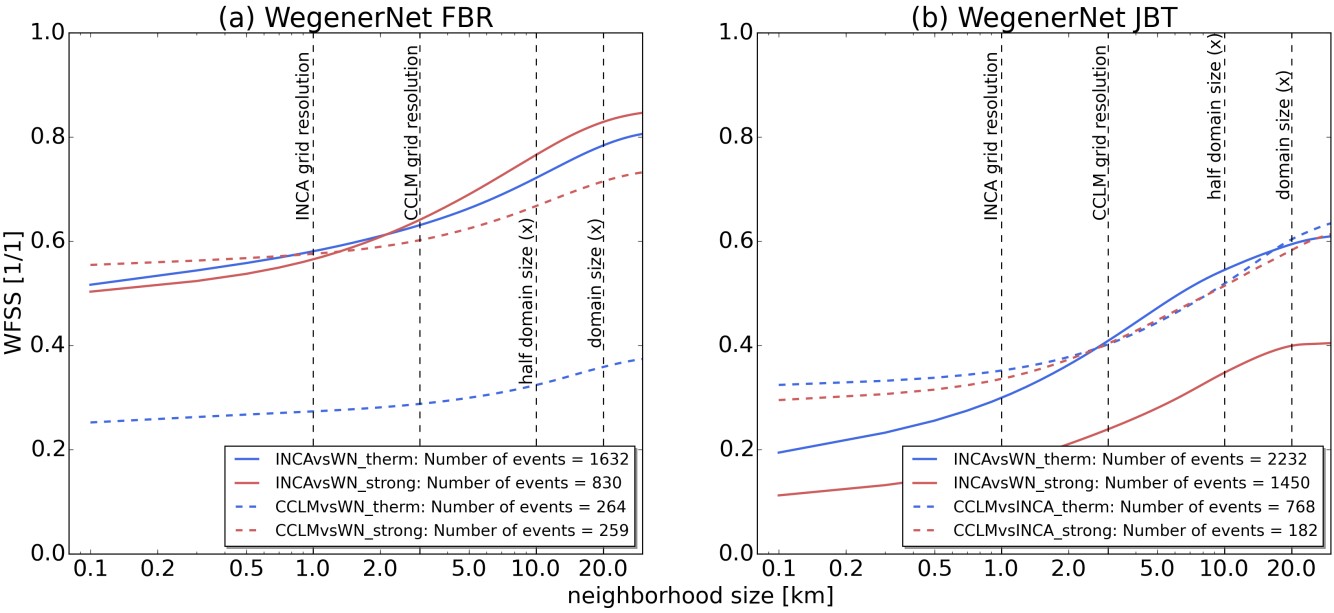

**Figure 3.** The event averaged Wind Fractions Skill Score (WFSS) results for the WegenerNet FBR (a), compared to the WegenerNet JBT (b), for the four defined evaluation cases in each region (see legend; indicating also the number of events included). See Table 1 for more information on the evaluation cases.

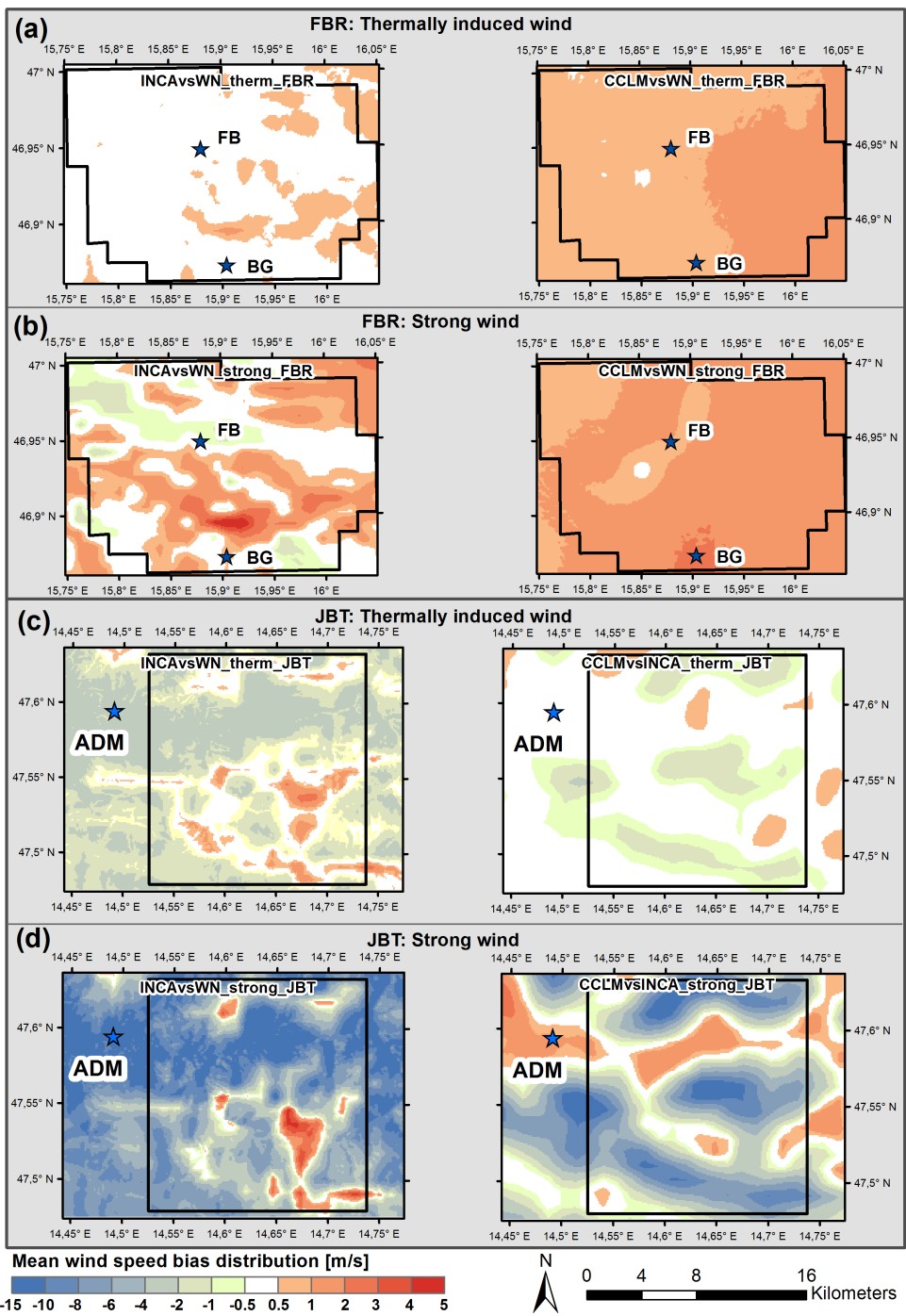

**Figure 4.** Mean wind speed bias distribution for the WegenertNet FBR (a, b) and the WegenerNet JBT (c, d): (a, b) INCA versus WegnerNet (left) and CCLM versus WegenerNet (right) for (a) thermally induced wind events and (b) strong wind events, and (c, d) WegenerNet versus INCA (left) and CCLM versus INCA (right) for (c) thermally induced wind events and (d) strong wind events.

**Table 1.** Characteristics of wind field evaluation cases used for the WegenerNet, INCA, and CCLM intercomparisons (top half for the WegenerNet FBR; bottom half for the WegenerNet JBT region).

| Evaluation Case | Type | Region | Modeled dataset | Reference dataset | Period |
|---|---|---|---|---|---|
| *WegenerNet FBR* | | | | | |
| INCAvsWN_therm_FBR | thermally | FBR | INCA | WN | 2008-2017 |
| INCAvsWN_strong_FBR | strong | FBR | INCA | WN | 2008-2017 |
| CCLMvsWN_therm_FBR | thermally | FBR | CCLM | WN | 2008-2010 |
| CCLMvsWN_strong_FBR | strong | FBR | CCLM | WN | 2008-2010 |
| *WegenerNet JBT* | | | | | |
| INCAvsWN_therm_JBT | thermally | JBT | INCA | WN | 2012-2017 |
| INCAvsWN_strong_JBT | strong | JBT | INCA | WN | 2012-2017 |
| CCLMvsINCA_therm_JBT | thermally | JBT | CCLM | INCA | 2008-2010 |
| CCLMvsINCA_strong_JBT | strong | JBT | CCLM | INCA | 2008-2010 |

**Table 2.** Limits for the selection of thermally induced or strong wind events for the defined evaluation cases shown in Table 1 (top half for the WegenerNet FBR; bottom half for the WegenerNet JBT).

| Evaluation Case | Variables[a] | $\overline{v}\,[\mathrm{m\,s^{-1}}]$ (reference data[b]) | $v\,[\mathrm{m\,s^{-1}}]$ (reference data[b]) | $rh\,[\%]$ (reference data[b]) | $Q_{g,m} - Q_{n(g),o}\,[\mathrm{W\,m^{-2}}]$ (reference data[b]) | $Q_n\,[\mathrm{W\,m^{-2}}]$ (reference data[b]) | Number of events[c] |
|---|---|---|---|---|---|---|---|
| | | | | *WegenerNet FBR* | | | |
| INCAvsWN_therm_FBR | | – | <1.5 (RS$_{dm}$) | <65.0 (RS$_{dm}$) | <100.0 (RS$_{dm}$) | <30.0 (RS$_{nm}$) | 1632 |
| INCAvsWN_strong_FBR | | >2.5 (WN$_{dm}$) | >3.0 (WN$_{hm}$) | – | – | – | 830 |
| CCLMvsWN_therm_FBR | | – | <1.5 (RS$_{dm}$) | <65.0 (RS$_{dm}$) | <100.0 (RS$_{dm}$) | <30.0 (RS$_{nm}$) | 264 |
| CCLMvsWN_strong_FBR | | >2.5 (WN$_{dm}$) | >3.0 (WN$_{hm}$) | – | – | – | 259 |
| | | | | *WegenerNet JBT* | | | |
| INCAvsWN_therm_JBT | | – | <2.0 (SCH$_{dm}$) | <65.0 (SCH$_{dm}$) | <20.0 (SCH$_{dm}$) | <30.0 (SCH$_{nm}$) | 2232 |
| INCAvsWN_strong_JBT | | >9.5 (WN$_{dm}$) | >9.0 (INCA$_{hm}$) | – | – | – | 1450 |
| CCLMvsINCA_therm_JBT | | – | – | <65.0 (WEI$_{dm}$) | <20.0 (WEI$_{dm}$) | – | 768 |
| CCLMvsINCA_strong_JBT | | >6.0 (INCA$_{dm}$) | >6.0 (INCA$_{hm}$) | – | – | – | 182 |

[a]$\overline{v}$: average wind speed; $v$: wind speed; $rh$: relative humidity; $Q_{g,m} - Q_{(n)g,o}$: difference between mean modeled global radiation ($Q_{g,m}$) and observed net radiation ($Q_{n,o}$) for the WegenerNet FBR and difference between $Q_{g,m}$ and observed global radiation ($Q_{g,o}$) for the WegenerNet JBT; $Q_n$: net radiation.

[b]RS: Reference Station (Nr. 77); SCH: Schroeckalm station; WEI: Weidendom station; dm: daytime mean value from observations (from sunrise till sunset); nm: nighttime mean value from observations (from sunset till sunrise); WN$_{dm}$: daily mean value from gridded WegenerNet wind speed; WN$_{hm}$: hourly mean value from gridded WegenerNet wind speed; INCA$_{dm}$: daily mean value from gridded INCA wind speed; INCA$_{hm}$: hourly mean value from gridded INCA wind speed.

[c]Hourly wind events; i.e., hours are used as the base period for the statistical analysis.

**Table 3.** Statistical performance parameters used for the intercomparison of the wind field modeling results.

| Parameter | Equation | Remarks |
|---|---|---|
| Wind fractions skill score | $WFSS = 1 - \dfrac{\sum_k \sum_{i,j} [O_k(i,j) - M_k(i,j)]^2}{\sum_k \left\{ \sum_{i,j} O_k(i,j)^2 + \sum_{i,j} M_k(i,j)^2 \right\}}$ | $O_k$: fraction values for observations for wind class $k$ at location $i,j$; $M_k$: fraction values for model data for wind class $k$ at location $i,j$ (Skok and Hladnik, 2018; Roberts, 2008) |
| Asymptotic WFSS | $AWFSS = 1 - \dfrac{\sum_k \left(f_k^O - f_k^M\right)^2}{\sum_k \left\{ \left(f_k^O\right)^2 + \left(f_k^M\right)^2 \right\}}$ | $f_k^O$: frequency of wind class $k$ in the observations; $f_k^M$: frequency of wind class $k$ in the model data (Skok and Hladnik, 2018) |
| Bias | $B = \frac{1}{N} \sum_{i=1}^{N} \left( v_{m,i} - v_{o,i} \right)$ | $v_m$: modeled wind speed; $v_o$: observed wind speed |
| Standard deviation of observed wind speed | $SD_o = \sqrt{\frac{1}{(N-1)} \sum_{i=1}^{N} (v_{o,i} - \overline{v_o})^2}$ | $v_o$: observed wind speed; $\overline{v_o}$: mean observed wind speed |
| Root-mean-square error | $RMSE = \sqrt{\frac{1}{N} \sum_{i=1}^{N} (v_{m,i} - v_{o,i})^2}$ | $v_m$: modeled wind speed; $v_o$: observed wind speed |
| Correlation coefficient | $R = \frac{1}{(N-1)} \sum_{i=1}^{N} \left( \frac{v_{m,i} - \overline{v_m}}{\sigma_m} \right) \left( \frac{v_{o,i} - \overline{v_o}}{\sigma_o} \right)$ | $v_m$: modeled wind speed; $\overline{v_m}$: mean modeled wind speed; $v_o$: observed wind speed; $\overline{v_o}$: mean observed wind speed; $\sigma_m$: standard deviation of modeled wind speed; $\sigma_o$: standard deviation of observed wind speed |
| Mean absolute error of wind direction | $MAE_{\mathrm{dir}} = \frac{1}{N} \sum_{i=1}^{N} \left\{ \arccos[\cos(\phi_{m,i} - \phi_{o,i})] \right\}$ | $\phi_m$: modeled wind direction; $\phi_o$: observed wind direction |

**Table 4.** Statistical performance measures calculated for the evaluation cases from Table 1, for the WegenerNet FBR and the WegenerNet JBT region. See Table 3 for more information on the calculation of the parameters.

| Evaluation Case | $AWFSS$ [1] | $B$ [$\mathrm{m\,s^{-1}}$] | $SD_o$ [$\mathrm{m\,s^{-1}}$] | $RMSE$ [1] | $R$ [1] | $MAE_{\mathrm{dir}}$ [°] |
|---|---|---|---|---|---|---|
| *WegenerNet FBR* | | | | | | |
| INCAvsWN_therm_FBR | 0.81 | 0.34 | 0.74 | 0.79 | 0.67 | 38 |
| INCAvsWN_strong_FBR | 0.85 | 0.50 | 1.04 | 1.66 | 0.34 | 14 |
| CCLMvsWN_therm_FBR | 0.38 | 1.32 | 0.72 | 1.85 | 0.37 | 55 |
| CCLMvsWN_strong_FBR | 0.74 | 1.01 | 1.03 | 1.28 | 0.57 | 14 |
| *Mean Value* | 0.70 | 0.79 | 0.88 | 1.40 | 0.49 | 30 |
| *WegenerNet JBT* | | | | | | |
| INCAvsWN_therm_JBT | 0.61 | -1.37 | 2.37 | 2.97 | 0.20 | 68 |
| INCAvsWN_strong_JBT | 0.39 | -6.69 | 3.97 | 8.60 | 0.16 | 39 |
| CCLMvsINCA_therm_JBT | 0.64 | -0.23 | 1.32 | 1.31 | 0.40 | 56 |
| CCLMvsINCA_strong_JBT | 0.63 | -3.79 | 4.24 | 5.52 | 0.08 | 25 |
| *Mean Value* | 0.57 | -3.04 | 2.98 | 4.62 | 0.20 | 47 |