# Peer review of "A spatial evaluation of high-resolution wind fields from empirical and dynamical modeling in hilly and mountainous terrain"

_Geoscientific Model Development, 2018_

## Referee Comment (RC1) · Anonymous Referee #1 · 30 Nov 2018

**Abstract**

This study comparatively evaluates three different wind products: a dataset of very high resolution (100 m) observations, an analysis product that blends modelled and observed fields (resolution about 1 km), and a pure regional climate model simulation (resolution 3 km). The comparison is carried out in two regions, labelled FBR and JDT, being the latter characterised by a more complex orography. The assessment is mostly based on a rather new metric, the Wind Fractional Skill Score (WFSS) that avoids the "double penalty problem". They separate the analysis into "calm" and "windy" events, and conclude that the 3 km spatial resolution of the regional climate model is not suf-

ficient to reproduce some of the characteristics of the wind field as recorded by the observations, especially in calm events. As expected, the agreement is better in the region characterised by less complex orography. Both, the analysis product as well as the climate model, tend to overestimate wind speed for strong wind events.

**General comment**

I think the authors make a good work at taking the rare opportunity provided by an unique grid of very highly resolved observations (both in time and space) to carry out a valuable comparison between products obtained through simulations and observations in two areas with markedly different orography. Further, it employs a new skill metric that cleverly combines the performance respect to wind speed and direction, and allows the evaluation of model performance at various degrees of neighbourhood. The text is, in my opinion, well written and easy to read, although it acknowledge that I am not native speaker. I have not found mayor flaws in the methodology and the way conclusions are drawn from the results. Still, I have found minor caveats or issues that perhaps deserve corrections or improved explanations before being considered for publication.

**Major comments**

As I said above, I do not find issues that deserve the category "major". Still, there are few issues that are more or less general and that is hard to allocate in a given line, as I do below with other minor comments.

1. The term "dynamical modelling" is repeated through the manuscript, and even in the title. I think this expression it is not very common in the Regional Climate

Modelling literature. This term seems to combine two more common expressions: "regional climate modelling" and "dynamical downscaling". Both are used in the literature more or less interchangeably, but I think "dynamical modelling" is not generally used. The reason for this is that, technically, a Global Circulation Model is also dynamical modelling, but I'm sure the authors do not mean this type of model. Therefore, I would advise to stick to one of the two aforementioned alternatives.

2. The authors refer to two former publications (Schlanger et al. 2017, 2018) where the WPG seems to be further described. I acknowledge that I didn't read these publications, but it is not clear to me what this article improves or how it complements the formers. I think putting emphasis somewhere in the introduction on what new issues/questions this new article tries to address, compared to the formers, would help to frame this work and to better justify why it is necessary.

3. The INCA dataset assimilates observations. Then this dataset is compared/validated with respect to the WPG, which are also observations. Are they the same? Are WPG observations assimilated to produce INCA? I assume not, as otherwise there would be an important circularity issue.

4. I'm not sure what is meant by a "wind event". I understand that the criteria in Table 2 is applied on an hourly basis, right? Are then the events hourly-based, i.e. a given hour might be included as a calm event, while the next one might be included as strong? Or do the authors select for instance the whole day when at least a single hour within the day meet the criteria? Another way of posing this question is, are there as many events as hours within each period?

5. Another detail I could not understand is how the WFSS is calculated for different spatial scales. Is the data interpolated onto successive grids with coarser resolution?

**Minor comments (in order of appearance)**

1. The abstract is in my opinion longer than necessary. For instance, between lines 5 and 10 a great amount of details are given about the datasets. This level of detail is overwhelming at this early point of the paper, and distracts the reader from the main conclusions of the manuscript.

2. Pag 2, Line 9: course-resolution → coarse-resolution.

3. Pag 2, Line 15: "data fusion". I think a more precise term is "data assimilation" or "assimilation of observations".

4. Pag 2, Line 19: "dynamical regional climate models" → "regional climate models".

5. Pag 3, Lines 3-8 These two paragraphs read as a summary of the methodology. I do not think this is necessary in the introduction.

6. Pag 3, Line 10: I was not aware of the concept "two penalty problem". Therefore I was puzzled to read this without either a reference or a couple of lines that briefly summarise what is the deal with this. It is explained later, so I would advise to bring those explanations already here.

7. Pag 4, Line 6: "eleven" → 11 (for consistency reasons with the way this is reported for FBR)

8. Pag 4: Lines 20-26: Is it really necessary this amount of detail about how the data about temperature and humidity is produced for this system, given that these fields are not used in the manuscript?

9. Pag 5, Lines 15-16: "Therefore the output shows errors in regions with low station density" The model resolution does not imply that there are larger errors in areas with low station density. Why would it be the case? The validation is more difficult, but it could be that the model does a good job. We just don't know.

10. Pag 5, Line 22: The number of vertical levels in the RCM (not only the driving dataset) is an important parameter worth to mention.

11. Pag 7, Line 23: the units (m s$^{-1}$) should not be italic. This applies to several locations through the manuscript. Please review them.

12. Pag 8, Line 15 says that wind speeds are systematically underestimated. This is curious, as normally models tend to overestimate wind speed. Indeed, in the conclusions (Page 13, Line 19) this is noted when it is stated that wind speed are overestimated in both types of events. Isn't this contradictory? Please clarify the details.

13. Page 10, Line 21: "fundamentally able". Do the authors mean "unable"?

14. A bottleneck of WFSS is that it does not allow to disentangle if low skill is driven by problems with wind speed or direction. However in Pag 10, from lines 29, this is somehow solved, and low skill is attributed to errors to these two variables separately. But it is not obvious how these conclusions can be drawn from the shown figures. Is this based on an analysis that is not shown in the manuscript?

15. Page 12, Line 21: where → were.

16. The conclusions are overly long. They review every single detail of the results and after reading them is not obvious what are the take-home messages. I advise to summarise the conclusions to leave the most important and general conclusions, those that can be exported to other studies/regions.

17. This may seem as a tiny detail, but the fact that the panels in Fig. 1 do not follow the expected order (a, then b, finally c) puzzled me for a couple of minutes until I realised that FBR (labelled b, and firstly described in the text) is actually the last panel of the figure. Perhaps a trivial re-ordering of the panels following a more intuitive order might facilitate the reading.

---

## Author Comment (AC1) · 22 Dec 2018

We thank Referee #1 for the valuable and constructive feedback to our manuscript. We carefully considered all comments and made due effort to account for the concerns expressed, and we think it really helped improving the readability and quality of the text and how we convey the findings. We also would like to thank the referee for the care related to remaining typos and spelling mistakes.

Responses to Major comments:

1) The term "dynamical modelling" is repeated through the manuscript, and even in the

title. I think this expression it is not very common in the Regional Climate Modelling literature. This term seems to combine two more common expressions: "regional climate modelling" and "dynamical downscaling". Both are used in the literature more or less interchangeably, but I think "dynamical modelling" is not generally used. The reason for this is that, technically, a Global Circulation Model is also dynamical modelling, but I'm sure the authors do not mean this type of model. Therefore, I would advise to stick to one of the two aforementioned alternatives.

Answer: Thank you for this hint, we will carefully recheck our usage of the general term "dynamical modeling" (in the sense of empirical modeling vs. dynamical modeling) and will replace it by a more specific term where we see needed, such as "regional climate modeling" or "dynamical high-resolution climate modeling" and so.

2) The authors refer to two former publications (Schlanger et al. 2017, 2018) where the WPG seems to be further described. I acknowledge that I didn't read these publications, but it is not clear to me what this article improves or how it complements the formers. I think putting emphasis somewhere in the introduction on what new issues/questions this new article tries to address, compared to the formers, would help to frame this work and to better justify why it is necessary.

Answer: OK, we agree that the introduction about the ongoing work described in this article in relation to the two former articles gains from more context. Hence we will insert a relevant paragraph in the introduction to clarify how this article complements the formers, as follows.

Inserted paragraph (page 2, after line 26): "... For both networks, diagnostic wind fields at a high resolution grid of 100 m × 100 m are generated by a weather diagnostic application, the WegenerNet Wind Product Generator (WPG). Schlager et al. (2017) introduced the WPG and its performance evaluation for the WegenerNet FBR, which was then advanced by Schlager et al. (2018) to the WegenerNet JBT region and a longer-term evaluation in both the FBR and JBT regions. Jointly these studies

established the level of quality of the empirical WPG wind fields. In this study, we now make use of these empirical high-resolution wind fields as reference data in order to intercompare them with empirical-dynamical wind field analysis data and ..."

3) The INCA dataset assimilates observations. Then this dataset is compared/validated with respect to the WPG, which are also observations. Are they the same? Are WPG observations assimilated to produce INCA? I assume not, as otherwise there would be an important circularity issue.

Answer: No, INCA does not assimilated WegenerNet data; and indeed we intentionally keep them independent just to avoid such circularity issues, yes. Having said this and based on rechecking our related description we agree, though, that the description of which station measurements are used in which model is a bit vague. We therefore will improve this a bit to make clear that observations from ZAMG stations (the ones used in the INCA analyses) are not used as model input for the WPG and vice versa that the INCA just uses observations from ZAMG stations, but not from WegenerNet stations as model input. We will do this as follows.

Improved description in Section 2.2 (starting on page 4, line 31): "... again with a maximum latency of about 1-2 hours. To keep the meteorological input data of the WPG independent from the data pertaining to the other operational station networks, observations from the ZAMG stations (violet stars in Fig.1a and Fig.1b) and other external stations are not used as WPG input. For the WegenerNet FBR, the gridded wind fields..."

Improved description in 2.3 INCA and COSMO-CLM data (starting on page 5, line 18): "In the WegenerNet FBR area, INCA assimilates observations from the ZAMG Feldbach and Bad Gleichenberg station (violet stars in Fig.1b) to the NWP's first guess, and within the WegenerNet JBT region, observations from the ZAMG Admont station are used. However, data from WegenerNet FBR and JBT stations are not used in INCA data assimilation and hence the WPG fields can be used for independent evaluation."

4) I'm not sure what is meant by a "wind event". I understand that the criteria in Table 2 is applied on an hourly basis, right? Are then the events hourly-based, i.e. a given hour might be included as a calm event, while the next one might be included as strong? Or do the authors select for instance the whole day when at least a single hour within the day meet the criteria? Another way of posing this question is, are there as many events as hours within each period?

Answer: Thanks for this comment which led us to notice that some further information regarding data selection would be helpful. To implement this, we will therefore modify some text passages in section 3.1 (Events for wind field evaluation). For example, the number of events shown in Table 2 corresponds to hourly events, but the data selection method for thermally induced wind events differs from the method for selecting strong wind events. In general, the data selection for thermally induced wind events is based on daytime and nighttime mean values as indicated by subscripts (dm, nm) in Table 2, which are also explained in the footnotes of this table. If a day is selected as autochthonous day, all 24 hours from this day are used for evaluating thermally induced wind events, i.e., such a day contributions 24 hourly wind events. In case of strong wind events, we compared the hourly mean values from the datasets with the hourly thresholds defined in Table 2 (hm subscripts). If the hourly mean value is larger than the defined threshold, this event is used for evaluating strong wind speeds. We will explicitly include a further line of footnote to Table 2, making clear that the "Number of events" column denotes hourly wind events as the basis for the statistical analysis, and we will modify the text as follows.

Modifications (starting page 6, line 18): "For the estimation of autochthonous days, we compared the observed daytime mean values of wind speed (v) and relative humidity (rh) as well as the nighttime mean values of net radiation (Qn) from selected stations with the respective thresholds...... If all criteria are fulfilled for a given day, the data from the entire day are added to the thermally induced wind events dataset, leading to 24 hourly events (i.e., 24 hourly-mean wind speed values)." And starting

page 6, line 31: "The strong wind events, caused by synoptic weather conditions such as cyclones and frontal system at larger scale, are selected on an hourly basis, by comparing hourly-mean values from the gridded reference datasets with defined minimum wind speeds (Table 2, v for strong wind speed cases). If the hourly-mean value of this reference dataset is larger than the defined minimum wind speed, the data of this reference dataset and the corresponding model dataset are used as part of the hourly event data for evaluating strong wind events."

5) Another detail I could not understand is how the WFSS is calculated for different spatial scales. Is the data interpolated onto successive grids with coarser resolution?

Answer: The calculation for different spatial scales is performed for defined neighborhood sizes, which have to be odd integers. A neighborhood size (n) defines the side length of a square, which is moved as sliding window over the dataset (e.g., n=5 corresponds to a neighborhoods size of 500 m at a spatial resolution of 100 m, and the square hence contains 25 grid points). We calculated the WFSS values for neighborhood sizes from n=1 to n=2N - 1, where N is the number of grid points of the largest domain size from the WegenerNet FBR or the WegenerNet JBT. The maximum domain size of 2N - 1 was used to ensure that the sliding window is large enough to always encompass the whole domain at every position - as a consequence, the fractions inside the domain are guaranteed to be the same at all locations within the domain and further enlarging the neighborhood will not change the WFSS value.

We will add additional text to section 3.2 (Statistical evaluation methods) to explain the calculation for different neighborhood sizes, as follows.

Additional text (starting at page 7 line 14) "...for different neighborhood sizes. The neighborhood sizes must be defined as odd integers. A neighborhood size of n defines a square consisting of n x n grid points, i.e., it denotes the side length of the square (e.g., for n=5 the square contains 25 grid points). These squares of defined neighborhood sizes are moved as sliding windows over the datasets, centered successively at

each grid point, whereby the area outside the domain is assumed to contain no wind class. In terms of FSS value, . . ."

Responses to Minor comments (in order of appearance):

1) The abstract is in my opinion longer than necessary. For instance, between lines 5 and 10 a great amount of details are given about the datasets. This level of detail is overwhelming at this early point of the paper, and distracts the reader from the main conclusions of the manuscript.

Answer: Ok, we agree that the abstract gives too detailed information. We will therefore reduce the level of detail regarding the explanation of the two meteorological station networks at the beginning and also somewhat the discussion of the results at the end of the abstract.

2) Pag 2, Line 9: course-resolution → coarse-resolution. Answer: Ok, will be done

3) Pag 2, Line 15: "data fusion". I think a more precise term is "data assimilation" or "assimilation of observations".

Answer: Ok, we will use "data assimilation" instead of "data fusion"

4) Pag 2, Line 19: "dynamical regional climate models" → "regional climate models".

Answer: Ok, will be done

5) Pag 3, Lines 3-8 These two paragraphs read as a summary of the methodology. I do not think this is necessary in the introduction.

Answer: Thank you for this hint. We agree, that this information is also given in section "3 Evaluation events and methods" and we will therefore remove these two paragraphs from the introduction.

6) Pag 3, Line 10: I was not aware of the concept "two penalty problem". Therefore I was puzzled to read this without either a reference or a couple of lines that briefly

summarise what is the deal with this. It is explained later, so I would advise to bring those explanations already here.

Answer: Thank you for this hint. We will move the explanation regarding the "double penalty" to this paragraph to immediately explain this kind of penalty. 7) Pag 4, Line 6: "eleven"→11 (for consistency reasons with the way this is reported for FBR)

Answer: Ok, will be done

8) Pag 4: Lines 20-26: Is it really necessary this amount of detail about how the data about temperature and humidity is produced for this system, given that these fields are not used in the manuscript?

Answer: Thank you for this hint, we agree that the gridded fields of temperature, precipitation, and relative humidity are not so relevant for this manuscript. We therefore will remove the (too) specific description parts about the lapse rate and the different interpolation methods for the generation of these fields.

9) Pag 5, Lines 15-16: "Therefore the output shows errors in regions with low station density" The model resolution does not imply that there are larger errors in areas with low station density. Why would it be the case? The validation is more difficult, but it could be that the model does a good job. We just don't know.

Answer: Thanks, we agree that this statement is not correct at this position of the text. We therefore will remove this sentence and modify the text in this paragraph to ensure, that the statement is related to the INCA analysis algorithm and not to the RCM's first guess. We will modify as follows.

Modified Text (page 5, line 16): "The INCA wind fields have already been evaluated for a moderately hilly region in the north of Austria (47.8° N–49° N, 13.8° E–17° E), where the wind analyses show significantly higher errors compared to the statistical results from other meteorological variables. These higher errors mainly root in the limited representativeness of station data, as well as on the low station density, which can be

[Figure]

only partly compensated by INCA's analysis algorithms (Haiden et al., 2011)."

10. Pag 5, Line 22: The number of vertical levels in the RCM (not only the driving dataset) is an important parameter worth to mention.

Answer: Ok, we will add additional text regarding the number of vertical levels. The COSMO-CLM simulations are provided for 40 vertical levels. The first level is simulated for 10 m above ground and the last level corresponds to the 100 hPa level, whereby the vertical resolution is higher for the boundary layer and decreases towards to the top level.

11) Pag 7, Line 23: the units (m s-1) should not be italic. This applies to several locations through the manuscript. Please review them.

Answer: Ok, will be corrected.

12) Pag 8, Line 15 says that wind speeds are systematically underestimated. This is curious, as normally models tend to overestimate wind speed. Indeed, in the conclusions (Page 13, Line 19) this is noted when it is stated that wind speed are overestimated in both types of events. Isn't this contradictory? Please clarify the details.

Answer: Thank for noticing this. The statement "systematically underestimated" is not fully correct in the context of what we try to address in the corresponding section (Pag 9, not Pag 8). In this section we are explaining the behavior of the WFSS for selected wind events and not for event-averaged statistical results. Therefore, the underestimation by the COSMO-CLM model explained in the text refers to a single event. We will correct the corresponding sentence as follows. Modified Text (page 9, line 13): "This discrepancy leads to the smallest WFSS values at all neighborhood sizes for this region (Fig. 2e, COSMOvsWN_strong_FBR) and indicates that the dynamically modeled COSMO wind speeds are underestimated relative to the empirically diagnosed wind speeds for this hourly event."

13) Page 10, Line 21: "fundamentally able". Do the authors mean "unable"?

[Figure]

Answer: Yes, we will change it to "unable"

14) A bottleneck of WFSS is that it does not allow to disentangle if low skill is driven by problems with wind speed or direction. However in Pag 10, from lines 29, this is somehow solved, and low skill is attributed to errors to these two variables separately. But it is not obvious how these conclusions can be drawn from the shown figures. Is this based on an analysis that is not shown in the manuscript?

Answer: It has to be noted that the WFSS can also be used to separately evaluate the two wind components, for example by classifying the datasets just based on wind direction. In general, the definition of the classes should reflect what a user wants to verify. We used the advantage of the WFSS and evaluated wind speed and wind direction in a combined way. Regarding the separate evaluation of both variables in relation to Fig. 3, we agree that this conclusion cannot be drawn by simply interpreting this figure and that additional information is needed. The behavior of the influence of wind speed or wind direction on the WFSS is indicated by the results of error measures additionally calculated by traditional statistical methods. These are summarized in Table 4 and the generated mean wind speed bias distribution map, illustrated in Fig. 4. Furthermore, we visually interpreted the windroses for most of the events (the windroses for all events are not shown in this manuscript, Fig. 2 just shows windroses for selected events, for good illustration). To make clear of how we draw this conclusion, we will modify the corresponding text passages and refer there to the results calculated by traditional methods. The spatial displacement and the biases for the INCAvsWN_therm_JBT case are mainly caused by the differences in wind directions for these thermally induced wind events, indicated by the large mean absolute error of wind direction (MAEdir) (Table 4). Modified text is as follows.

Modified text (page 10, line 32): "The INCAvsWN_strong_JBT case shows the lowest values at all neighborhood sizes, but this time caused by the differences in the wind speed categories. These low values are caused by the INCA-analyzed wind speeds, which, in case of strong winds, are overestimated in the summit regions and underes-

[Figure]

Interactive
comment

timated in the valley regions. Slightly overestimated WegenerNet wind speeds in the Enns valley are somewhat reinforcing the difference between the INCA and the WegenerNet wind speeds. These differences in wind speed especially in the valley and the summit regions become obvious from Fig. 4c and 4d and are discussed in further detail below."

15) Page 12, Line 21: where→were.

Answer: Ok, will be done

16) The conclusions are overly long. They review every single detail of the results and after reading them is not obvious what are the take-home messages. I advise to summarise the conclusions to leave the most important and general conclusions, those that can be exported to other studies/regions

Answer: Thank you for your advice, we agree that the conclusion gives too detailed information, which especially applies to the discussion of the results. We will therefore summarize the explanation of the results and shortly discuss what's relevant for ongoing next steps of work and other studies/regions.

17) This may seem as a tiny detail, but the fact that the panels in Fig. 1 do not follow the expected order (a, then b, finally c) puzzled me for a couple of minutes until I realised that FBR (labelled b, and firstly described in the text) is actually the last panel of the figure. Perhaps a trivial re-ordering of the panels following a more intuitive order might facilitate the reading.

Answer: Thank you, we agree that the panel sequence and the corresponding labeling is a bit confusing. We will therefore move the FBR panel to the top of Fig. 1 and label it with (a), and the JBT panel to the bottom and label it with (b). Furthermore, the overview in the middle of Fig. 1 will be denoted as middle panel in the text; the discuss-panels is (a) and (b) and so everything will be clear.

We thank Refereee #1 again very much for the valuable comments that will help to

improve the manuscript.

---

## Referee Comment (RC2) · Anonymous Referee #2 · 3 May 2019

Review of: A spatial evaluation of high-resolution wind fields from empirical and dynamical modeling in hilly and mountainous terrain

*Summary*

This paper describes an analysis of surface winds (not sure at what height though??) in 2 Austrian mountain regions, one with relatively simple topography and another with more complex detailed topography. The analysis compares a re-gridded product from in-situ observations of wind (and other) observations to provide a regularly gridded fields, an empirical analysis type product and a regional climate model. The analysis employs a number of statistical measures to determine the performance of the empirical analysis and climate model in two objectively determined wind regimes 'thermal' events and 'strong wind' events. In general, performance with either model is poorer over the more complex terrain and for thermal events.

In general, this is a reasonable descriptive paper of standard measures of performance for wind speed simulations. However, I do feel like there is insufficient detail on 2 particular fronts of the paper. Firstly, it feels that the modeling approaches and the CALMET regridding are just presented as is, with no critical discussions of the pros and cons of the methodologies and how they could affect the analysis here. The COSMO model in particular is somewhat of a mystery and there is no speculation as to what the model may be doing wrong to have poorer performance, beyond just saying it is not high enough resolution (even though 1 to 3 km is not that big of a jump). Given the different behavior of the two regimes, the question that sparks most for me is that maybe COSMO is poorer at simulating the wind profiles of 'thermal events' versus 'strong wind events'. This is particularly pertinent to the study since the conclusions are that we need more observations and no evidence is shown that we may need better models. Thermal events are potentially complex interplays between differential heating and turbulence, which ultimately lead to the wind profile and yet none of the thermodynamic (or even wind) structures are examined from the model to understand this. So, in general an elaboration of the models' shortcomings is needed and more interpretation beyond just a description of the comparison, as this will inform model improvements which I presume is the end goal here.

Minor comments Abstract 1,14: 'skill scores' 1,16-18: I found the ordering of this confusing. 1,24: Even if the thermal events are 'strong events'? 2,3: What do you mean by decent? Acceptable?

1. Introduction 2,7: What's the definition of surface wind here – 10m? 2,9: This is potentially possible it just won't be high resolution. And how does it hamper interpolation? 2,28: Are the WegenerNet fields used as part of the INCA analysis and to also validate INCA? 3,7: Given you are referring to COLSMO-CLM as a climate model, I am unsure

how to think of actual synoptically overlapping periods with WNet? 3.12: 'and provide'

2. Study Areas and Model Data 3.26: Sensitive in that it has already experience change? 3.30: Could elaborate a bit her. Katabatic winds, turbulent PBL,... 4.10: Are not both regions subject to synoptic weather conditions given their close proximity? 4.22: Are there dangers in interpolating both relative humidity and temperature separately since one is a non-linear function of the other, due to saturation temperature being a non-linear function of T? 4.28: What are the meteorological fields used? Does this actually include explicit wind observations and what vertical levels are used? 5.25-29: This is a little confusing here. Do you mean the COSMO model is driven continuously by ECMWF on the domain boundaries for 2008-2010, and you are describing the time stepping numerics? Also, what are setting 'based on shallow convection'?

3. Evaluation Events and Methods

6.11: 'autochthounous' I had to look this up! But I am still not sure what is being referred to. 6.10-15: Is there any presumption of diurnal variations here? 6.20: 'daily global radiation'? surface solar? 6.30: These 'thermal wind events' have not really been defined yet. 6.31-34: Is this the only criteria for the 'strong wind events'. Given it is large scale synoptic would it be more meaningful to have an area coherence footprint or temporal longevity criteria. 7.7: Is this to reduce penalty in both space and time? Fig 2: This is very confusing indeed. Are these just snapshots of a particular day, even a specific hour, given the time stamp at the top of each plot? 7.33-35: I do not understand this at all. 'Ensemble of events'?? 8.21: You're implying here that Alpine pumping is a local phenomenon that arises due to local forcings topography. However, wouldn't you expect a model to do well at this if it is simply forced by the analyzed wind at its boundaries? 8.34: But wasn't this the less challenging terrain compared to the other region? 9.1-10: Although the wind roses do give a good summary of the biases in wind direction, the key thing to understanding the differences of course is the synoptic distribution over the domain. This shows that INCA is not southerly enough mostly in the southern part of the domain. Is this explainable from this perspective? 9.11-12:

This is really surprising given that COSMO is all yellow/orange whereas the other fields are seeing weaker speed values in the greens. 9.16: 1th?? 9.24-29: This needs more interpretation here. What aspect of the dynamical model is failing? Is it the solution itself or is it the synoptic setup? Why does 8/1/2012 mostly succeed but this day fail? 10.6: Won't this always be true of COSMO in these synoptic circumstances? However, the scale of the features for the high wind regions here are actually above the coarser grid scales of COSMO, so this lack of resolution reasoning is not correct is it? 10.21: Unable instead of able?? 10.25: Again, though isn't this the simpler terrain region? Fig 4: It is very surprising that the COSMO model has a widespread systematic bias over the simpler FBR region, but a much reduced systematic bias in general over the much more complex terrain of the JBT region.
* * *

---

## Author Comment (AC2) · 23 May 2019

We thank Referee #2 very much for the valuable and quite detailed feedback to our manuscript. We carefully considered all comments and made due effort to account for the concerns expressed; and we think it really helped improving the readability and quality of the text and how we convey the findings.

Respones to Major comments:

1) Firstly, it feels that the modeling approaches and the CALMET regridding are just presented as is, with no critical discussions of the pros and cons of the methodologies

and how they could affect the analysis here.

Answer: Thank you for this hint, we reconsidered the description about advantages and disadvantages of the different modeling approaches. With regard to the empirical modeling approach, we referred only to former publications and agree, that additional information on this modeling approach should be given in the text. Also the description about the INCA and the CCLM (we will use, for simplicity, CCLM instead of COSMO-CLM from now on) model needs to be improved, especially with regard to internal numerical settings and the lateral boundary. We will therefore add additional text to the model data sections 2.2 and 2.3.

With regard to the CALMET re-gridding, the CALMET-based wind fields were not re-sampled in order to avoid information losses in these high-resolution data. The coarser INCA and CCLM data were resampled and mapped onto the high-resolution WPG grid. In addition, we have performed sensitivity tests for different interpolation methods and found no significant changes in the statistical results. (See paragraph on page 5 from line 31 to 34). We reconsidered also our description related to this; we think that this particular description about the re-gridding of the data is already detailed enough.

[revised manuscript text omitted]

keeps the modeled synoptic patterns in agreement with the observed ones."

2) The COSMO model in particular is somewhat of a mystery and there is no speculation as to what the model may be doing wrong to have poorer performance, beyond just saying it is not high enough resolution (even though 1 to 3 km is not that big of a jump). Given the different behavior of the two regimes, the question that sparks most for me is that may be COSMO is poorer at simulating the wind profiles of 'thermal events' versus' strong wind events'. This is particularly pertinent to the study since the conclusions are that we need more observations and no evidence is shown that we may need better models. Thermal events are potentially complex interplays between differential heating and turbulence, which ultimately lead to the wind profile and yet none of the thermodynamic (or even wind) structures are examined from the model to understand this. So, in general an elaboration of the models' shortcomings is needed and more interpretation beyond just a description of the comparison, as this will inform model improvements which I presume is the end goal here.

Answer: We agree, that these flow patterns are influenced by complex interplays of thermodynamic structures. The model behavior of CCLM is also very complex and disentangling the various influences would far exceed the scope of this study. Therefore, at this point, we can only come up with more speculative interpretations. Based on recent discussions with our internal RCM experts and with ZAMG model developers, we have come to the conclusion that the main argument for medium-term model improvements lies indeed in higher-spatial-resolution simulations. What has not yet been mentioned in the manuscript is that the CCLM model uses an advection scheme, which causes additional smoothing of the terrain. The scheme is implemented to avoid numerical instability, but its diffusion damping causes an effective resolution, which is quite lower than in the INCA model. This ultimately leads to quite low spatial variability in the CCLM wind fields and may explain the high uncertainty in the modelled wind directions, especially under weak synoptic forcing. In addition, flow patterns may significantly divert from the observations. Due to the orographic smoothing, flow-over

patterns occur more frequently than flow-around patterns. However, if flow-over patterns occur more frequently, the influence of the orographic speed-up effect (Taylor et al., 1987) becomes more dominant. In contrast, if mountains and hills are higher, more flow-around patterns and flow-splitting patterns occur, which are favoring even negative orographic speed-up effects (Hewer, 1998). This might be the reason for the overestimation of the wind speed and its improvement under strong wind conditions in FBR. In JBT, however, the underrepresentation of the orography becomes even more striking. The central mountain in this region in CCLM is about 500 m lower than in INCA. This gives a severe deformation of the CCLM wind field and clearly indicates the requirement for improving the treatment of orography in high-resolution simulations. In principal, the ALARO model suffers from similar shortcomings. However, since the model's output is corrected with the help of station data, the wind fields in INCA are much better in agreement with WegenerNet data than CCLM. Beside higher-resolution simulations, improvements in the CCLM can be expected from using a newly developed advection scheme that allows to circumvent the horizontal diffusive damping. If actually higher-resolution models were evaluated, however, the topographic shading through the terrain becomes increasingly important, especially for the simulation of thermally induced wind events. Such methods are not implemented in the ALARO and were switched off in the CCLM model for the generation of the data used in this study. Other influences on wind are: (1) misleading land cover properties (e.g., of the roughness lengths), (2) underestimation of land cover heterogeneity, (3) the negligence of the so-called zero-plane displacement (Oke, 2009), and (4) no use of a 3D turbulence parameterization, based for example on large eddy simulations. We will address these model limitations and possible improvements in the text as follows:

Additional text starting on page 11, line 30, will read: "...study area (right panel of Figure 4a). The overestimation of the wind speeds for the WegenerNet FBR can be explained by the too frequent flow-over patterns simulated for this region, which lead to a more dominant orographic speed-up effect. Due to the orographic smoothing, flow-over patterns are generally more frequent than flow-around patterns, especially for the

WegenerNet FBR with its small differences in altitude (Taylor et al., 1987)."

Additional text starting on page 11, line 35, will read: "...fields in this case. These large B-values are probably also due to the speed-up effect explained for the above case CCLMvsWN_therm_FBR." Additional text starting on page 12, line 15, will read: "...explained above. The negative B-values are likely caused by negative orographic speed-up effects, which are preferred in flow-around patterns and flow-splitting patterns, which occur especially when the differences in the altitude of ridges of mountains are large." Additional text starting on page 13, line 16, will read: "...dataset from this model. Although the difference in the numerical resolution between INCA (1 km grid spacing) and CCLM (3 km grid spacing) is only a factor of 3, CCLM is not able to resolve small-scale wind patterns. This occurs for multiple reasons: 1) due to the 3rd-order advection scheme with its horizontal diffusion damping, the effective resolution in CCLM is several times coarser than the numeric grid spacing (Ogaja and Will, 2016); 2) the orography is smoothed as well, so that individual mountain ridge and valley structures are removed. For example, the mountain peak of the Hochtor with its 2396 m elevation in the center of the WegenerNet JBT region is lowered by about 500 m in the CCLM model. Modified text starting on page 13, line 35, will read: "...terrain, and the limited effective resolution of 10 times the numeric grid spacing of 3 km x 3 km." Additional text starting on page 13, line 35, will read: "...CCLM. Improvements can be expected from latest developments in the numerical core of CCLM by Ogaja and Will (2016): they have enabled an improvement of the effective resolution by a factor of 2 via introducing a 4th-order advection scheme that allows to circumvent the horizontal diffusion damping." Additional text starting on page 14, line 5, will read: "...INCA-analyzed wind fields. At higher-resolutions, the topographic shading through the terrain becomes increasingly important, especially for the simulation of thermally induced wind events. Such methods have not yet been implemented into the ALARO model, but may help to generate more realistic wind fields in the future." Modified text starting on page 14, line 6, will read: "...wind fields and the application of the new 4th-order advection scheme from Ogaja and Will (2016) in a convection-permitting configuration would also be a

promising route for further investigations of how this may improve the modeling of wind patterns in a complex terrain."

Responses to Minor comments:

Abstract:

1) 1.14: 'skill scores': Answer: Ok, will be done

2) 1.16-18: I found the ordering of this confusing: Answer: Thank you for this hint. We considered to change ordering of the description related to the model intercomparisons; but to be consistent with the defined evaluation cases (INCAvsWN_xxxx_FBR, CCLMvsWN_xxxx_FBR; see Table 1, we preferred it's better to keep the existing ordering in the text (see text between INCA and WegenerNet than between CCLM and WegenerNet wind fields).

3) 1.24: Even if the thermal events are 'strong events'?

Answer: A criterion for selecting a day as autochthonous day, which includes thermally induced wind events is generally weak wind speeds (see Table 2). Therefore, the sample of strong wind events in the thermally induced cases is too small, and no statement can be made as to whether a model is better for such strong events under autochthonous weather conditions. Specifically, CCLMvsWN_therm_FBR does not contain strong wind events, INCAvsWN_therm_FBR contains seven strong wind events, CCLMvsINCA_therm_JBT does not contain strong wind events, and for the INCAvsWN_therm_JBT case we estimated just 16 strong events.

4) 2.3: What do you mean by decent? Acceptable?

Answer: Yes, ok, we will change to 'acceptable'

1. Introduction:

5) 2.7: What's the definition of surface wind here – 10 m?

Answer: This statement refers to the first levels within the PBL, which are influenced by the terrain.

6) 2.9: This is potentially possible it just won't be high resolution. And how does it hamper interpolation?

Answer: Thank you for this hint; with this statement we refer to high-resolution wind field modeling on a regional to local scale. To make clear that the generation of realistic high-resolution wind fields is not possible with coarse-resolution models or by an interpolation of wind station data, we will modify the text as follows:

Modified text: "Therefore, realistic high-resolution wind fields cannot be generated with coarse-resolution models or by a simple interpolation of wind station data onto regular grids."

7) 2.28: Are the WegenerNet fields used as part of the INCA analysis and to also validate INCA?

Answer: No, to avoid circularity issues INCA does not use any WegenerNet data and vice versa no INCA data are used in the WPG. Due to the vague description of which data are used in which model and a comment from Referee 1, we will improve the text related to this.

Modified text: Please see point 3 in the document for the response to Referee #1 (https://editor.copernicus.org/index.php/gmd-2018-238-AC1.pdf?_mdl=msover_md&_jrl=365&_lcm=oc108lcm109w&_acm=get_comm_file&_ms=71779&c=154013&salt=156112

8) 3.7: Given you are referring to COLSMO-CLM as a climate model, I am unsure how to think of actual synoptically overlapping periods with WNet?

The COSMO model in climate mode implements several new features compared to the original COSMO weather model. For example, the vegetation state of soil is not assumed to be constant, or it is able to use not only initial values but also dynamic boundary data. The CCLM simulations where generated during the course of a previous study and cover the period Jan.2006 - Dec. 2009, and they were constrained at synoptic scale by assimilated ECMWF IFS fields – see the improved and more detailed CCLM description now included (answer to main comments above).

9) 3.12: 'and provide'

Ok, will change to provide

2. Study Areas and Model Data:

10) 3.26: Sensitive in that it has already experience change?

Yes, in this region climate change is already measurable. For example observational based studies show a strong summer temperature trend of 0.7 °C per decade (Kabas et al. 2011, Hohmann et al.2018).

11) 3.30: Could elaborate a bit her. Katabatic winds, turbulent PBL,...

Answer: Thank you for this hint; we will add additional text as follows and use the term "drainage wind" to refer to small-scale flows:

Additional text (starting at page 3 line 31) will read: "... (Lugauer and Winkler, 2005). Furthermore, nocturnal drainage winds, which are leading to cold air pockets, are relevant for this region, which is dominated by agriculture. Especially in fall and winter, the nocturnal cold air production is amplified by temperature inversions in relation to high-pressure weather conditions. In WegenerNet FBR, hillside locations are thermally preferred to valley locations at night."

12) 4.10: Are not both regions subject to synoptic weather conditions given their close proximity?

Answer: Yes, both regions are subject to synoptic weather conditions. With "westerly-flow synoptic weather conditions" we refer to general weather conditions, which lead to airflows with prevailing westerly wind directions and strong wind speeds at higher altitudes in the WegenerNet JBT. In the WegenerNet FBR, the damping effect of the

thermal stratification on synoptic winds is larger, which cause a low amplitude between the month with the average strongest winds and the month with the average lowest winds.

13) 4.22: Are there dangers in interpolating both relative humidity and temperature separately since one is a non-linear function of the other, due to saturation temperature being a non-linear function of T?

Answer: Thank you for this comment. The gridded fields of temperature, precipitation, and relative humidity are not used as model input (which uses station data) and are therefore not relevant for this manuscript. For this reason, and because of a comment from Referee 1, we will remove the description parts about how these fields are generated (please see also point 8 in the response to Referee #1).

14) 4.28: What are the meteorological fields used? Does this actually include explicit wind observations and what vertical levels are used?

Answer: The main purpose of the generated meteorological fields is to investigate weather and climate as well as evaluating RCMs (please see Page 1, lines 23-26). Yes, the CALMET model used in the WPG generates mean wind fields based on observed wind speed and wind direction from the WegnerNet stations, among other needs. The INCA system assimilates data from the ZAMG stations. In this study, we are using the mean wind fields at 10 m height for the model intercomparisions. We've rechecked the manuscript related to this, and noticed, that this important information of which height level is used for the model intercomparisons was missing, so we will therefore add additional text as follows.

Additional text (starting on page 4, line 32) will read: "...WegenerNet JBT starting in 2012. In this study, the wind fields at 10 m height level are used for the model intercomparisons." And starting on page 5, line 32, we will insert: "Furthermore, we resampled the wind fields at 10 m height level from these two models..."

15) 5.25-29: This is a little confusing here. Do you mean the COSMO model is driven continuously by ECMWF on the domain boundaries for 2008-2010, and you are describing the time stepping numerics? Also, what are setting 'based on shallow convection'?

For detailed information about numerical settings and driving data please see now point 2) in the responses to major comments above, where we provide now a quite more detailed model description.

3. Evaluation Events and Methods:

16) 6.11: 'autochthounous' I had to look this up! But I am still not sure what is being referred to.

Weather conditions that are determined by local or regional daily variations in temperature or pressure are referred to as autochthonous conditions. Such conditions are mostly caused in cases of low synoptic influences, by anti-cyclonic weather conditions and favors thermally induced flows.

17) 6.10-15: Is there any presumption of diurnal variations here?

The selection of autochthonous days is based only on the comparison of daytime and nighttime averages and no assumptions were made regarding daily variations (see page 6, lines 17-29 and Table 2). The results of this method show good agreement with another study, where such days have been manually selected (Oberth, U., 2010: Untersuchung der lokalen Windsysteme im Raum Feldbach unter besonderer Berücksichtigung von Kaltluftabflüssen. (in German). Master theses, 146 pp. [Available online at http://www.wegenernet.org/misc/MA_Oberth_2010_WegernerNet_Wind.pdf].)

18) 6.20: 'daily global radiation'? surface solar?

Depending on the region, we used the observed global radiation or net radiation as input for the selection method (See paragraph on page 6 from line 17 to 26 and Table 2).

[Figure]

19) 6.30: These 'thermal wind events' have not really been defined yet.

Answer: Thank you for this hint, we re-checked the description and will add additional text to ensure what is meant by thermally induced wind events.

The additional text (starting at page 6 line 11) will read: "...temperature and pressure gradients. These small-scale gradients lead to characteristic interacting systems of air motion, like slope winds and mountain-valley winds, and create complex everyday flow patterns. The autochthonous days..."

20) 6.31-34: Is this the only criteria for the 'strong wind events'. Given it is large scale synoptic would it be more meaningful to have an area coherence footprint or temporal longevity criteria.

Answer: Thank you for this hint; we have noticed that important information on another criterion is missing both in the text and in Table 2.

In order to determine the weather situation during prolonged weather with strong winds, the respective days were selected on the basis of the daily average wind speed. Subsequently we have chosen the hourly events from these days. We will therefore add additional text to the corresponding paragraph. Furthermore, we will add the additional limit-values for the selection of these days to Table 2. And yes, further improvements in the selection of such days can be expected through the use of e.g. longevity criteria or frontal detection methods, but these were not applied during the course of this study since considered beyond the scope of due efforts to this end.

Additional text (starting at page 6, line 31) will read: "The strong wind events, caused by synoptic weather conditions such as cyclones and frontal system at larger scale, are selected on an hourly basis from preselected days, by comparing hourly mean values from gridded reference datasets with defined minimum wind speeds. These preselected days were estimated by comparing the daily average wind speed from the gridded datasets with a defined minimum average wind speed (Table 2, "v (with

overline)" and "v" for strong wind speed cases)."

21) 7.7: Is this to reduce penalty in both space and time?

No, the FSS is a spatial and not a spatiotemporal verification metric.

4 Results:

22) Fig 2: This is very confusing indeed. Are these just snapshots of a particular day, even a specific hour, given the time stamp at the top of each plot?

Answer: Yes, this Figure illustrates just single one-hour events, indicated by the hourly period at the top of each plot. We agree that especially the labeling of the hourly periods is somewhat confusing. We will therefore improve this labeling (will be changed for example from 7/29/2009 04:00:00 PM-17:00:00 to 29.07.2009 16:00:00-17:00:00). Furthermore, we have adapted the color map for the representation of the three wind classes from the windroses to the one of the ten classes from the wind fields.

23) 7.33-35: I do not understand this at all. 'Ensemble of events'??

Answer: The event-averaged score values are calculated based on averaging the one-hour event WFSS values over all the hourly events for a specific case. In this study we calculated eight event-averaged score values which are shown in Figure 3. To make this clear, we will modify the accompanying text. Moreover, we have noticed that we refer to this value as case-averaged score value here (page 7 line 32) and as event-averaged score value in all other parts of the text. We will now uniformly refer to this parameter as event-averaged score value.

The modification (starting page 7, line 32) will read: "...selected events. The event-averaged score values are calculated based on averaging these one-hour event WFSS values over all the hourly events within the analyzed multi-year period, for each evaluation case listed in Table 1."

24) 8.21: You're implying here that Alpine pumping is a local phenomenon that arises

due to local forcings topography. However, wouldn't you expect a model to do well at this if it is simply forced by the analyzed wind at its boundaries?

Answer: Thank you for this hint; we imply here that Alpine pumping is a regional, not a local phenomenon. In contrast to thermally induced local winds, this phenomenon leads to compensating flows on a regional scale, which are called Randgebirgswind and its counterpart, the Antirandgebirgswind. Especially in case of autochthonous weather conditions, the Antirandgebirgswind is influencing the WegenerNet FBR in the afternoon. In our former studies we have evaluated the WPG also for such conditions and found good results, which is mainly due to the dense station network with wind observations. Due to the fact that alpine pumping is a very complex process and INCA has only two station observations available in the WegenerNet FBR, we did not expect any specific results about the quality of the simulated wind fields for such conditions. The analyses of the INCA fields shows, that INCA is able to adequately simulate the significant wind pattern of the Antirandgebirgswind, which affects not only the ridges of the hills but also the valleys in the WegenerNet FBR.

We will add additional information on the spatial scale of the Antirandgebirgswind to the corresponding text passage.

The additional text (starting page 7, line 32) will read: "...with maximum wind speeds of about 2.5 m s$-1$. The Antirandgebirgswind is a compensating flow between the bordering mountains of the eastern Alps, and the hilly country region of southeastern Styria (called Riedelland), which is comprising the WegenerNet FBR (Wakonigg, 1978)."

25) 8.34: But wasn't this the less challenging terrain compared to the other region?

Answer: In general, in this section we describe the characteristics of example wind fields and the model results for representative hourly events for each evaluation case, first for the WegenerNet FBR and then for the WegenerNet JBT. The results of each individual evaluation case are described and then compared with them from a corresponding case within the same region. In this specific paragraph we describe the results for the CCLMvsWN_therm_FBR case, which are then compared with the IN-CAvsWN_therm_FBR case. Both cases are defined for thermally induced wind events, which correspond to the WegenerNet FBR. We have rechecked this paragraph and recognized, that corresponding evaluation case definition for the CCLM case is not mentioned in the text and will therefore modify the text as follows:

The modified text (page 8, line 32) will read: "...to INCA evaluation cases, which indicates a large bias (Fig.2 CCLMvsWN_therm_FBR)."

26) 9.1-10: Although the wind roses do give a good summary of the biases in wind direction, the key thing to understanding the differences of course is the synoptic distribution over the domain. This shows that INCA is not southerly enough mostly in the southern part of the domain. Is this explainable from this perspective?

We agree that the wind roses in combination with the wind fields give a good intuitive notion how well the INCA wind field matched the WegenerNet field. In this specific example, the large AWFSS and therefore small bias in wind classes over the whole domain is not reflected by the wind classification result shown in the windroses. In this example we are trying to show the advantage of calculating the WFSS based on an azimuthal class rotation (for explanation of class rotation please see page 8 lines 1-10). A calculation of the WFSS without rotating the classes would lead to a poor AWFSS of about 0.6 (instead of >0.97). We also agree, that the low WFSS at small neighborhood sizes is mainly caused by the differences in wind sectors, especially in the southern part of the domain. Furthermore, parts of the area differ in wind speed classes. To illustrate this in the text, we will add information to the corresponding paragraph.

The additional text (page 9, line 7) will read: "... (INCAvsWN_strong_FBR). These low WFSS values are mainly caused by the differences in wind direction classes, especially in the southern part of the domain and through some spatial displacements in wind speed classes."

27) 9.11-12: This is really surprising given that COSMO is all yellow/orange whereas the other fields are seeing weaker speed values in the greens.

Answer: Thank you for this hint. In this particular case, we indeed (inadvertently) used the wrong wind speed limits to create the wind rose. We also re-checked the code for the calculation of the WFSS and could confirm the correct limits are implemented here. We will adjust the lower-middle panel of Fig. 2b and the corresponding text in the manuscript.

The modified text (page 9, line 7) will read: "Regarding the CCLM data (lower-middle panel of Fig. 2b), the whole wind field shows wind speeds from about 6.5 m s-1 to 7.5 m s-1 and is therefore assigned to the wind class with wind speeds higher than 6 m s-1. Whereas, for the WegenerNet wind fields, a large proportion is assigned to the class with wind speeds from 3 m s-1 to 6 m s-1 of this region (Fig. 2e,CCLMvsWN_strong_FBR) and indicates that the dynamically modeled CCLM wind speeds are systematically overestimated relative to the empirically diagnosed wind speeds."

28) 9.16: 1th??

Answer: Yes, on the 1st of August 2012 and on the 31st of May 2008 the winds in the WegenerNet JBT were thermally driven (see Fig. 2c).

29) 9.24-29: This needs more interpretation here. What aspect of the dynamical model is failing? Is it the solution itself or is it the synoptic setup? Why does 8/1/2012 mostly succeed but this day fail?

Answer: Also in this case, we mainly attribute the uniform wind directions simulated with the CCLM to the too strongly smoothed terrain in the model. For such events under low synoptic forcing, both wind fields show a too low spatial variability in wind direction. Regarding wind speed, the INCA wind field shows some variability with higher wind speeds in parts of the summit regions compared to wind speeds at lower altitude.

Furthermore, a valley wind in the Enns valley becomes obvious. Probably the analysis part of the INCA model leads to a somewhat better representation of the wind field. We will add additional text to draw attention to such effects. Could you please indicate what you mean with "Why does 8/1/2012 mostly succeed but this day fail"? We checked through the text but were not sure what's meant. For this CCLMvsINCA_therm_JBT event we are analyzing the 31th of Mai 2008 from 13:00-14:00.

30) Additional text (starting page 9 line 25): "…(bottom right panels of Fig. 2c). Especially in the CCLM, the smoothed terrain leads to uniform wind speeds and directions. Regarding the INCA wind fields, some variability in wind speed, with higher values in the summit regions and lower values at lower altitudes in the valleys of this region, can be observed. Furthermore, a valley wind in the Enns valley is simulated by INCA. Probably the analysis part of the INCA model with its higher-resolved DEM and assimilated ZAMG observations leads to a somewhat better representations of the wind field. The shift in wind directions between CCLM and INCA leads to low WFSS values for all neighborhood sizes, including the lowest asymptotic value of all examples, indicating a very poor representation of the wind field by the dynamical modeling of the CCLM in this challenging mountainous terrain."

31) 10.6: Won't this always be true of COSMO in these synoptic circumstances? However, the scale of the features for the high wind regions here are actually above the coarser grid scales of COSMO, so this lack of resolution reasoning is not correct is it?

Answer: For the WegenerNet FBR the wind fields are systematically overestimated which become obvious in the CCLMvsWN_strong_FBR case and in the statistical evaluation results (cf. also Fig. 4). For the WegenerNet JBT the low wind speeds are probably explained by negative orographic speed-up effects (Hewer, 1998) caused by a too smoothed terrain, compared to the WegnerNet FBR, where speed-up effects are leading to stronger wind speeds. For a detailed information about this speed-up effects see also point 2) in the responses to Major comments above. We will add additional text about such effects to the 4.2 Statistical evaluation results section (for this text see

also point 34) below, which deals with a similar question).

32) 10.21: Unable instead of able??

Answer: Yes, ok, will change to unable

33) 10.25: Again, though isn't this the simpler terrain region?

Yes, here we describe the performance of the CCLM in comparison to the INCA model for strong wind speeds for the hilly WegenerNet FBR. The influence of the terrain (e.g. channeling of air flow through the valleys) on the synoptic flow field is smaller in this hilly region than in the WegenerNet JBT region. That's why the CCLM shows similar performance as the INCA model despite the lower resolution for this region.

34) Fig 4: It is very surprising that the COSMO model has a widespread systematic bias over the simpler FBR region, but a much reduced systematic bias in general over the much more complex terrain of the JBT region.

Answer: Thank you for this hint; the difference in these bias values between the two regions is probably again attributed to the speed up effects. For more information please see point 2) in the response to Major comments above. Furthermore, it has to be noted that in comparison to the WegenerNet FBR region the INCA data and not the WegenerNet data were used as reference for the evaluation of the CCLM, due to missing WegenerNet data. Since the CCLM wind fields show small bias values for thermally induced wind events, compared to the INCA wind fields, similar results as in the CCLMvsINCA_therm_JBT case can be expected for a comparison of CCLM with WegenerNet data. In case of strong wind events, the intercomparison of the CCLM with the INCA model shows opposite patterns than the INCAvsWN_strong_JBT case, but with smaller bias values. Therefore the same bias values in attenuated form are to be expected for a comparison of the CCLM with WegenerNet data. We will add this information to the text as follows.

The additional text starting on page 12, line 6, will read: "... values can be observed

for this case. Due to these small bias values, similar results as for this can be expected for a comparison of CCLM with WegenerNet data." And additional text starting on page 12, line 15, will read: "...The negative B values are probably attributable to negative orographic speed-up effects (Hewer, 1998), which are favored in case of flow-around patterns and flow-splitting patterns, which especially occur if the ridges of mountains and hills are higher."

---

## Author Response (AR1)

**Response to Reviewer #1 and Reviewer #2**

of "A spatial evaluation of high-resolution wind fields from empirical and dynamical modeling in hilly and mountainous terrain"
by C Schlager, G Kirchengast, J Fuchsberger, Alexander Kann, and Heimo Truhetz.
Submitted to GMD, October 2018;

*We thank the Reviewers again very much for the valuable and quite detailed feedback to our manuscript. We carefully considered all comments and made due effort to account for the concerns expressed; and we think it really helped improving the comprehensibility and quality of the text and how we convey the findings. We also would like to thank the Reviewers for the care also related to remaining typos and spelling mistakes. We corrected in line with all of these suggestions.*
*Comments by the Reviewer are* black upright*, our responses blue italic. (Page and line numbers used in our responses below refer to the revised manuscript; to make this clear they are quoted like "now p10 L20-25")*

**Response to Reviewer # 1 from interactive discussion**

**Answerers to your Major comments:**

1) The term "dynamical modelling" is repeated through the manuscript, and even in the title. I think this expression it is not very common in the Regional Climate Modelling literature. This term seems to combine two more common expressions: "regional climate modelling" and "dynamical downscaling". Both are used in the literature more or less interchangeably, but I think "dynamical modelling" is not generally used. The reason for this is that, technically, a Global Circulation Model is also dynamical modelling, but I'm sure the authors do not mean this type of model. Therefore, I would advise to stick to one of the two aforementioned alternatives.
*Answer: Thank you for this hint, we carefully rechecked our usage of the general term "dynamical modeling" (in the sense of empirical modeling vs. dynamical modeling) and replaced it by a more specific term where we see needed, such as "regional climatemodeling" or "dynamical high-resolution climate modeling" and so (now p1 L3, p1 L18, p2 L12, p2 L28, p2 L31, p10 L22).*

2) The authors refer to two former publications (Schlanger et al. 2017, 2018) wherethe WPG seems to be further described. I acknowledge that I didn't read these publications, but it is not clear to me what this article improves or how it complements the formers. I think putting emphasis somewhere in the introduction on what new is-sues/questions this new article tries to address, compared to the formers, would help to frame this work and to better justify why it is necessary.
*Answer: OK, we agree that the introduction about the ongoing work described in this article in relation to the two former articles gains from more context. Therefore, we have included a relevant paragraph in the introduction to clarify how this article complements the formers (now p2 L23-28).*

3) The INCA dataset assimilates observations. Then this dataset is compared/validated with respect to the WPG, which are also observations. Are they the same? Are WPG observations assimilated to produce INCA? I assume not, as otherwise there would be an important circularity issue:

*Answer: No, INCA does not assimilated WegenerNet data; and indeed we intentionally keep them independent just to avoid such circularity issues, yes. Having said this and based on rechecking our related description we agree, though, that the description of which station measurements are used in which model is a bit vague. We therefore improved this a bit to make clear that observations from ZAMG stations (the ones used in the INCA analyses) are not used as model input for the WPG and vice versa that the INCA just uses observations from ZAMG stations, but not from WegenerNet stations as model input (now p4 L29-31, p6 L13-16).*

4) I'm not sure what is meant by a "wind event". I understand that the criteria in Table 2 is applied on an hourly basis, right? Are then the events hourly-based, i.e. a given hour might be included as a calm event, while the next one might be included as strong? Or do the authors select for instance the whole day when at least a single hour within the day meet the criteria? Another way of posing this question is, are there as many events as hours within each period.

*Answer: Thanks for this comment which led us to notice that some further information regarding data selection would be helpful. To implement this, we modified some text passages in section 3.1 (Events for wind field evaluation). For example, the number of events shown in Table 2 corresponds to hourly events, but the data selection method for thermally induced wind events differs from the method for selecting strong wind events. In general, the data selection for thermally induced wind events is based on daytime and nighttime mean values as indicated by subscripts (dm, nm) in Table 2, which are also explained in the footnotes of this table. If a day was selected as autochthonous day, all 24 hours from this day were used for evaluating thermally induced wind events, i.e., such a day contributions 24 hourly wind events. In case of strong wind events, we compared the hourly mean values from the datasets with the hourly thresholds defined in Table 2 (hm subscripts). If the hourly mean value is larger than the defined threshold, this event is used for evaluating strong wind speeds. We explicitly included a further line of footnote to Table 2, making clear that the "Number of events" column denotes hourly wind events as the basis for the statistical analysis, and modified the text (now p8 L2-4, p8 L17-22).*

5) Another detail I could not understand is how the WFSS is calculated for different spatial scales. Is the data interpolated onto successive grids with coarser resolution?

*Answer: The calculation for different spatial scales is performed for defined neighbor-hood sizes, which have to be odd integers. A neighborhood size (n) defines the side length of a square, which is moved as sliding window over the dataset (e.g., n=5 corresponds to a neighborhoods size of 500 m at a spatial resolution of 100 m, and the square hence contains 25 grid points). We calculated the WFSS values for neighbor-hood sizes from n=1 to n=2N - 1, where N is the number of grid points of the largest domain size from the WegenerNet FBR or the WegenerNet JBT. The maximum domains size of 2N - 1 was used to ensure that the sliding window is large enough to always encompass the whole domain at every position - as a consequence, the fractions inside the domain are guaranteed to be the same at all locations within the domain and further enlarging the neighborhood will not change the WFSS value. We added additional text to section 3.2 (Statistical evaluation methods) to explain the calculation for different neighborhood sizes (now p8 L33-34, p9 L1-3).*

**Answerers to your Minor comments:**

1) The abstract is in my opinion longer than necessary. For instance, between lines 5 and 10 a great amount of details are given about the datasets. This level of detail is overwhelming at this early point of the paper, and distracts the reader from the main conclusions of the manuscript

*Answer: Ok, we agree that the abstract gives too detailed information. We therefore reduced the level of detail regarding the explanation of the two meteorological station networks at the beginning and also somewhat the discussion of the results at the end of the abstract (now p1 L7-8, P1 L21-25).*

2) Pag 2, Line 9: course-resolution→coarse-resolution
*Answer: OK, done (now p2 L5)*

3) Pag 2, Line 15: "data fusion". I think a more precise term is "data assimilation" or" assimilation of observations".
*Answer: Ok, we now use "data assimilation" instead of "data fusion (now p2 L11)*

4) Pag 2, Line 19: "dynamical regional climate models"→"regional climate models".
*Answer: OK, done (now p2 L12)*

5) Pag 3, Lines 3-8 These two paragraphs read as a summary of the methodology. I do not think this is necessary in the introduction.
*Answer: Thank you for this hint. We agree, that this information is also given in section "3 Evaluation events and methods" and we therefore removed these two paragraphs from the introduction.*

6) Pag 3, Line 10: I was not aware of the concept "two penalty problem". Therefore I was puzzled to read this without either a reference or a couple of lines that briefly summarise what is the deal with this. It is explained later, so I would advise to bring those explanations already here.
*Answer: Thank you for this hint. We moved the explanation regarding the "double penalty" to this paragraph to immediately explain this kind of penalty (now p3 L5-9).*

7) Pag 4, Line 6:"eleven"→11 (for consistency reasons with the way this is reported for FBR)
*Answer: OK, done (now p4 L6).*

8) Pag 4: Lines 20-26: Is it really necessary this amount of detail about how the data about temperature and humidity is produced for this system, given that these fields are not used in the manuscript?
*Answer: Thank you for this hint, we agree that the gridded fields of temperature, precipitation, and relative humidity are not so relevant for this manuscript. We therefore removed the (too) specific description parts about the lapse rate and the different interpolation methods for the generation of these fields.*

9) Pag 5, Lines 15-16: "Therefore the output shows errors in regions with low station density" The model resolution does not imply that there are larger errors in areas with low station

density. Why would it be the case? The validation is more difficult, but it could be that the model does a good job. We just don't know.

*Answer: Thanks, we agree that this statement is not correct at this position of the text. We therefore removed this sentence and modified the text in this paragraph to ensure, that the statement is related to the INCA analysis algorithm and not to the RCM's first guess (now p6 L11-12).*

10) Pag 5, Line 22: The number of vertical levels in the RCM (not only the driving dataset) is an important parameter worth to mention.

*Answer: The COSMO-CLM simulations are provided for 40 vertical levels. The first level is simulated for 10 m above ground and the last level corresponds to the 100 hPa level, whereby the vertical resolution is higher for the boundary layer and decreases towards to the top level. Based on your comment and a note of Referee 2 we now give detailed information about the model characteristics of all models (see also 2) from the Responses to Referee #2: under the Major comments) (WN: p4 L33-34, p5 L1-8), (INCA: now p5 L26-33, p6 L1-8), (CCLM: now p6 L27-34, p7 L1-14).*

11) Pag 7, Line 23: the units (m s-1) should not be italic. This applies to several locations through the manuscript. Please review them.

*Answer: Ok, corrected (now p9 L11, p10 L10, p10 L26, p11 L1, p11 L2, p11 L3, p13 L14, p13 L15, p13 L25, p13 L26, p13 L27, p13 L34, p14 L12).*

12) Pag 8, Line 15 says that wind speeds are systematically underestimated. This is curious, as normally models tend to overestimate wind speed. Indeed, in the conclusions (Page 13, Line 19) this is noted when it is stated that wind speed are overestimated in both types of events. Isn't this contradictory? Please clarify the details.

*Answer: Thank for noticing this. The statement "systematically underestimated" is not fully correct in the context of what we try to address in the corresponding section (Pag 9, not Pag 8). In this section we are explaining the behavior of the WFSS for selected wind events and not for event-averaged statistical results. Therefore, the underestimation by the COSMO-CLM model explained in the text refers to a single event. We corrected the corresponding sentence (now p11 L6-8).*

13) Page 10, Line 21: "fundamentally able". Do the authors mean "unable"?

*Answer: Yes changed it to "unable" (now p12 L18).*

14) A bottleneck of WFSS is that it does not allow to disentangle if low skill is driven by problems with wind speed or direction. However in Pag 10, from lines 29, this is somehow solved, and low skill is attributed to errors to these two variables separately. But it is not obvious how these conclusions can be drawn from the shown figures. Is this based on an analysis that is not shown in the manuscript?

*Answer: It has to be noted that the WFSS can also be used to separately evaluate the two wind components, for example by classifying the datasets just based on wind direction. In general, the definition of the classes should reflect what a user wants to verify. We used the advantage of the WFSS and evaluated wind speed and wind direction in a combined way. Regarding the separate evaluation of both variables in relation to Fig. 3, we agree that this conclusion cannot be drawn by simply interpreting this figure and that additional information*

*is needed. The behavior of the influence of wind speed or wind direction on the WFSS is indicated by the results of error measures additionally calculated by traditional statistical methods. These are summarized in Table 4 and the generated mean wind speed bias distribution map, illustrated in Fig.4. Furthermore, we visually interpreted the windroses for most of the events (the windroses for all events are not shown in this manuscript, Fig. 2 just shows windroses for selected events, for good illustration). To make clear of how we draw this conclusion, we modified the corresponding text passages and refer there to the results calculated by traditional methods. The spatial displacement and the biases for the INCAvsWN_therm_JBT case are mainly caused by the differences in wind directions for these thermally induced wind events, indicated by the large mean absolute error of wind direction (MAEdir) (Table 4) sentence (now p12 L32-34).*

15) Page 12, Line 21: where→were
*Answer: Ok, done (now p14 L26).*

16) The conclusions are overly long. They review every single detail of the results and after reading them is not obvious what are the take-home messages. I advise to summarise the conclusions to leave the most important and general conclusions, those that can be exported to other studies/regions.
*Answer: Thank you for your advice, we agree that the conclusion gives too detailed information, which especially applies to the discussion of the results. We therefore summarized the explanation of the results and shortly discuss what's relevant for ongoing next steps of work and other studies/regions. In addition, we now provide additional information on possible model improvements (now p15 L19-26, p16 L1-6, p16 L10-12).*

17) This may seem as a tiny detail, but the fact that the panels in Fig. 1 do not follow the expected order (a, then b, finally c) puzzled me for a couple of minutes until I realized that FBR (labelled b, and firstly described in the text) is actually the last panel of the figure. Perhaps a trivial re-ordering of the panels following a more intuitive order might facilitate the reading.
*Answer: Thank you, we agree that the panel sequence and the corresponding labeling is a bit confusing. We therefore moved the FBR panel to the top of Fig. 1 and labeled it with (a), and the JBT panel to the bottom and labeled it with (b). Furthermore, the overview in the middle of Fig. 1 is denoted as middle panel in the text; the discuss-panels is (a) and (b) and so everything is clear.*

**Response to Reviewer # 2 from interactive discussion**

**Answerers to your Major comments:**

1) Firstly, it feels that the modeling approaches and the CALMET regridding are just presented as is, with no critical discussions of the pros and cons of the methodologies and how they could affect the analysis here.

*Answer: Thank you for this hint, we reconsidered the description about advantages and disadvantages of the different modeling approaches. With regard to the empirical modeling approach, we referred only to former publications and agree, that additional information on this modeling approach should be given in the text. Also the description about the INCA and the CCLM (we now use CCLM instead of COSMO-CLM for the sake of simplicity) model needed to be improved, especially with regard to internal numerical settings and the lateral boundary. We therefore added additional text to the model data sections 2.2, 2.2 and 2.3 (now p4 L33-34, p5 L1-8, p5 L26-33, p6 L1-8, p6 L27-33, p7 L1-14).*

*With regard to the CALMET re-gridding, the CALMET-based wind fields were not resampled in order to avoid information losses in these high-resolution data. The coarser INCA and CCLM data were resampled and mapped onto the high-resolution WPG grid. In addition, we have performed sensitivity tests for different interpolation methods and found no significant changes in the statistical results. (See paragraph on page 5 from line 31 to 34). We reconsidered also our description related to this; we think that this particular description about the re-gridding of the data is already detailed enough (now p6 L17-20, p7 L15-17).*

2) The COSMO model in particular is somewhat of a mystery and there is no speculation as to what the model may be doing wrong to have poorer performance, beyond just saying it is not high enough resolution (even though 1 to 3 km is not that big of a jump). Given the different behavior of the two regimes, the question that sparks most for me is that may be COSMO is poorer at simulating the wind profiles of 'thermal events' versus' strong wind events'. This is particularly pertinent to the study since the conclusions are that we need more observations and no evidence is shown that we may need better models. Thermal events are potentially complex interplays between differential heating and turbulence, which ultimately lead to the wind profile and yet none of the thermodynamic (or even wind) structures are examined from the model to understand this. So, in general an elaboration of the models' shortcomings is needed and more interpretation beyond just a description of the comparison, as this will inform model improvements which I presume is the end goal here.

*Answer: We agree, that these flow patterns are influenced by complex interplays of thermodynamic structures. The model behavior of CCLM is also very complex and disentangling the various influences would far exceed the scope of this study. Therefore, at this point, we can only come up with more speculative interpretations.*

*Based on recent discussions with our internal RCM experts and with ZAMG model developers, we have come to the conclusion that the main argument for medium-term model improvements lies indeed in higher-spatial-resolution simulations. What has not yet been mentioned in the manuscript is that the CCLM model uses an advection scheme, which causes additional smoothing of the terrain. The scheme is implemented to avoid numerical instability, but its diffusion damping causes an effective resolution, which is quite lower than in the INCA model. This ultimately leads to quite low spatial variability in the CCLM wind fields and may explain the high uncertainty in the modelled wind directions, especially under weak synoptic forcing. In addition, flow patterns may significantly divert from the observations. Due to the orographic smoothing, flow-over patterns occur more frequently*

*than flow-around patterns. However, if flow-over patterns occur more frequently, the influence of the orographic speed-up effect (Taylor et al., 1987) becomes more dominant. In contrast, if mountains and hills are higher, more flow-around patterns and flow-splitting patterns occur, which are favoring even negative orographic speed-up effects (Hewer, 1998). This might be the reason for the overestimation of the wind speed and its improvement under strong wind conditions in FBR. In JBT, however, the underrepresentation of the orography becomes even more striking. The central mountain in this region in CCLM is about 500 m lower than in INCA. This gives a severe deformation of the CCLM wind field and clearly indicates the requirement for improving the treatment of orography in high-resolution simulations. In principal, the ALARO model suffers from similar shortcomings. However, since the model's output is corrected with the help of station data, the wind fields in INCA are much better in agreement with WegenerNet data than CCLM.*

*Beside higher-resolution simulations, improvements in the CCLM can be expected from using a newly developed advection scheme that allows to circumvent the horizontal diffusive damping. If actually higher-resolution models were evaluated, however, the topographic shading through the terrain becomes increasingly important, especially for the simulation of thermally induced wind events. Such methods are not implemented in the ALARO and were switched off in the CCLM model for the generation of the data used in this study.*

*Other influences on wind are: (1) misleading land cover properties (e.g., of the roughness lengths), (2) underestimation of land cover heterogeneity, (3) the negligence of the so-called zero-plane displacement (Oke, 2009), and (4) no use of a 3D turbulence parameterization, based for example on large eddy simulations.*

*We address now these model limitations and possible improvements in the text (now p13 L28-31, p14 L1-2, p14 L18-20, p15 L20-26, p16 L3-6, p16 10-16).*

**Answerers to your Minor comments:**

*Abstract:*

1 1.14: 'skill scores':
*Answer: Ok, done (now p1 L12).*

2 1.14: 1.16-18: I found the ordering of this confusing:
*Answer: Thank you for this hint. We considered to change ordering of the description related to the model intercomparisons; but to be consistent with the defined evaluation cases (INCAvsWN_xxxx_FBR, CCLMvsWN_xxxx_FBR; see Table 1, we preferred it's better to keep the existing ordering in the text (see text between INCA and WegenerNet than between CCLM and WegenerNet wind fields).*

31.14: 1.24: Even if the thermal events are 'strong events'?:
*Answer: A criterion for selecting a day as autochthonous day, which includes thermally induced wind events is generally weak wind speeds (see Table 2). Therefore, the sample of strong wind events in the thermally induced cases is too small, and no statement can be made as to whether a model is better for such strong events under autochthonous weather conditions. Specifically, CCLMvsWN_therm_FBR does not contain strong wind events, INCAvsWN_therm_FBR contains seven strong wind events, CCLMvsINCA_therm_JBT does*

*not contain strong wind events, and for the INCAvsWN_therm_JBT case we estimated just 16 strong events.*

**1. Introduction:**

5) 2.7: What's the definition of surface wind here – 10 m?
*Answer: This statement refers to the first levels within the PBL, which are influenced by the terrain.*

6) 2.9: This is potentially possible it just won't be high resolution. And how does it hamper interpolation?
*Answer: Thank you for this hint; with this statement we refer to high-resolution wind field modeling on a regional to local scale. To make clear that the generation of **realistic** high-resolution wind fields is not possible with coarse-resolution models or by an interpolation of wind station data, we modified the text (now p2 L4-6).*

7) 2.28: Are the WegenerNet fields used as part of the INCA analysis and to also validate INCA?
*Answer: No, to avoid circularity issues INCA does not use any WegenerNet data and vice versa no INCA data are used in the WPG. Due to the vague description of which data are used in which model and a comment from Referee 1, improved the text related to this (now p4 L29-31, p6 L13-16)*

8) 3.7: Given you are referring to COLSMO-CLM as a climate model, I am unsure how to think of actual synoptically overlapping periods with WNet?
*The COSMO model in climate mode implements several new features compared to the original COSMO weather model. For example, the vegetation state of soil is not assumed to be constant, or it is able to use not only initial values but also dynamic boundary data. The CCLM simulations where generated during the course of a previous study and cover the period Jan.2006 - Dec. 2009, and they were constrained at synoptic scale by assimilated ECMWF IFS fields – we improved and added more detailed CCLM description (for page and line references see point 2 under Major comments).*

9) 3.12: 'and provide'
*Ok, will change to "and provide" (now p3 L9).*

**2. Study Areas and Model Data:**

10) 3.26: Sensitive in that it has already experience change?
*Yes, in this region climate change is already measurable. For example observational based studies show a strong summer temperature trend of 0.7 °C per decade (Kabas et al. 2011, Hohmann et al.2018).*

11) 3.30: Could elaborate a bit her. Katabatic winds, turbulent PBL,...
*Answer: Thank you for this hint; we added additional text and use now the term "drainage wind" to refer to small-scale flows (now p3 L28-31).*

12) 4.10: Are not both regions subject to synoptic weather conditions given their close proximity?
*Answer: Yes, both regions are subject to synoptic weather conditions. With "westerly-flow synoptic weather conditions" we refer to general weather conditions, which lead to airflows with prevailing westerly wind directions and strong wind speeds at higher altitudes in the WegenerNet JBT. In the WegenerNet FBR, the damping effect of the thermal stratification on synoptic winds is larger, which cause a low amplitude between the month with the average strongest winds and the month with the average lowest winds.*

13) 4.22: Are there dangers in interpolating both relative humidity and temperature separately since one is a non-linear function of the other, due to saturation temperature being a non-linear function of T?
*Answer: Thank you for this comment. The gridded fields of temperature, precipitation, and relative humidity are not used as model input (which uses station data) and are therefore not relevant for this manuscript. For this reason, and because of a comment from Referee 1, we removed the description parts about how these fields are generated (please see also point 8 in the response to Referee #1).*

14) 4.28: What are the meteorological fields used? Does this actually include explicit wind observations and what vertical levels are used?
*Answer: The main purpose of the generated meteorological fields is to investigate weather and climate as well as evaluating RCMs (please see Page 1, lines 23-26).*
*Yes, the CALMET model used in the WPG generates mean wind fields based on observed wind speed and wind direction from the WegnerNet stations, among other needs. The INCA system assimilates data from the ZAMG stations.*
*In this study, we are using the mean wind fields at 10 m height for the model intercomparisions. We've rechecked the manuscript related to this, and noticed, that this important information of which height level is used for the model intercomparisons was missing, so we therefore added additional text related to this to the manuscript (now p4 L32, p, p6 L18, p7 L15).*

15) 5.25-29: This is a little confusing here. Do you mean the COSMO model is driven continuously by ECMWF on the domain boundaries for 2008-2010, and you are describing the time stepping numerics? Also, what are setting 'based on shallow convection'?
*Answer: For detailed information about numerical settings and driving data see point 2) in the responses to major comments above, where we also give the page and line numbers of the new text passages for a more detailed model description.*

**3. Evaluation Events and Methods:**

16) 6.11: 'autochthounous' I had to look this up! But I am still not sure what is being referred to.
*Answer: Weather conditions that are determined by local or regional daily variations in temperature or pressure are referred to as autochthonous conditions. Such conditions are mostly caused in cases of low synoptic influences, by anti-cyclonic weather conditions and favors thermally induced flows.*

17) 6.10-15: Is there any presumption of diurnal variations here?
*Answer: The selection of autochthonous days is based only on the comparison of daytime and nighttime averages and no assumptions were made regarding daily variations (see page 6, lines 17-29 and Table 2). The results of this method show good agreement with another study, where such days have been manually selected (Oberth, U., 2010: Untersuchung der lokalen Windsysteme im Raum Feldbach unter besonderer Berücksichtigung von Kaltluftabflüssen. (in German). Master theses, 146 pp. [Available online at http://www.wegenernet.org/misc/MA_Oberth_2010_WegernerNet_Wind.pdf].)*

18) 6.20: 'daily global radiation'? surface solar?
*Answer: Depending on the region, we used the observed global radiation or net radiation as input for the selection method (See paragraph on page 6 from line 17 to 26 and Table 2).*

19) 6.30: These 'thermal wind events' have not really been defined yet.
*Answer: Thank you for this hint, we re-checked the description and added additional text to ensure what is meant by thermally induced wind events (now p7 L27-28).*

20) 6.31-34: Is this the only criteria for the 'strong wind events'. Given it is large scale synoptic would it be more meaningful to have an area coherence footprint or temporal longevity criteria.
*Answer: Thank you for this hint; we have noticed that important information on another criterion is missing both in the text and in Table 2.*
*In order to determine the weather situation during prolonged weather with strong winds, the respective days were selected on the basis of the daily average wind speed. Subsequently we have chosen the hourly events from these days. We therefore added additional text to the corresponding paragraph. Furthermore, we added the additional limit-values for the selection of these days to Table 2.*
*And yes, further improvements in the selection of such days can be expected through the use of e.g. longevity criteria or frontal detection methods, but these were not applied during the course of this study since considered beyond the scope of due efforts to this end.*

21) 7.7: Is this to reduce penalty in both space and time?
*Answer: No, the FSS is a spatial and not a spatiotemporal verification metric.*

**4 Results:**

22) Fig 2: This is very confusing indeed. Are these just snapshots of a particular day, even a specific hour, given the time stamp at the top of each plot?
*Answer: Yes, this Figure illustrates just single one-hour events, indicated by the hourly period at the top of each plot. We agree that especially the labeling of the hourly periods is somewhat confusing. We improved this labeling (changed for example from 7/29/2009 04:00:00 PM-17:00:00 to 29.07.2009 16:00:00-17:00:00). Furthermore, we have adapted the color map for the representation of the three wind classes from the windroses to the one of the ten classes from the wind fields.*

23) 7.33-35: I do not understand this at all. 'Ensemble of events'??

*Answer: The event-averaged score values are calculated based on averaging the one-hour event WFSS values over all the hourly events for a specific case. In this study we calculated eight event-averaged score values which are shown in Figure 3. To make this clear, we modified the accompanying text. Moreover, we have noticed that we refer to this value as case-averaged score value here (page 7 line 32) and as event-averaged score value in all other parts of the text. We now uniformly refer to this parameter as event-averaged score value.*

24) 8.21: You're implying here that Alpine pumping is a local phenomenon that arises due to local forcings topography. However, wouldn't you expect a model to do well at this if it is simply forced by the analyzed wind at its boundaries?
*Answer: Thank you for this hint; we imply here that Alpine pumping is a regional, not a local phenomenon. In contrast to thermally induced local winds, this phenomenon leads to compensating flows on a regional scale, which are called Randgebirgswind and its counterpart, the Antirandgebirgswind. Especially in case of autochthonous weather conditions, the Antirandgebirgswind is influencing the WegenerNet FBR in the afternoon. In our former studies we have evaluated the WPG also for such conditions and found good results, which is mainly due to the dense station network with wind observations. Due to the fact that alpine pumping is a very complex process and INCA has only two station observations available in the WegenerNet FBR, we did not expect any specific results about the quality of the simulated wind fields for such conditions. The analyses of the INCA fields shows, that INCA is able to adequately simulate the significant wind pattern of the Antirandgebirgswind, which affects not only the ridges of the hills but also the valleys in the WegenerNet FBR.*

*We added additional information on the spatial scale of the Antirandgebirgswind to the corresponding text passage (now p10 L10-12).*

25) 8.34: But wasn't this the less challenging terrain compared to the other region?
*Answer: In general, in this section we describe the characteristics of example wind fields and the model results for representative hourly events for each evaluation case, first for the WegenerNet FBR and then for the WegenerNet JBT. The results of each individual evaluation case are described and then compared with them from a corresponding case within the same region. In this specific paragraph we describe the results for the CCLMvsWN_therm_FBR case, which are then compared with the INCAvsWN_therm_FBR case. Both cases are defined for thermally induced wind events, which correspond to the WegenerNet FBR. We have rechecked this paragraph and recognized, that corresponding evaluation case definition for the CCLM case is not mentioned in the text and added therefore additional text (now p10 L22).*

26) 9.1-10: Although the wind roses do give a good summary of the biases in wind direction, the key thing to understanding the differences of course is the synoptic distribution over the domain. This shows that INCA is not southerly enough mostly in the southern part of the domain. Is this explainable from this perspective?
*Answer: We agree that the wind roses in combination with the wind fields give a good intuitive notion how well the INCA wind field matched the WegenerNet field. In this specific example, the large AWFSS and therefore small bias in wind classes over the whole domain is not reflected by the wind classification result shown in the windroses. In this example we*

*are trying to show the advantage of calculating the WFSS based on an azimuthal class rotation (for explanation of class rotation please see page 8 lines 1-10). A calculation of the WFSS without rotating the classes would lead to a poor AWFSS of about 0.6 (instead of >0.97).*
*We also agree, that the low WFSS at small neighborhood sizes is mainly caused by the differences in wind sectors, especially in the southern part of the domain. Furthermore, parts of the area differ in wind speed classes. To illustrate this in the text, we added information to the corresponding paragraph (now p10 L30-31).*

27) 9.11-12: This is really surprising given that COSMO is all yellow/orange whereas the other fields are seeing weaker speed values in the greens.
*Answer: Thank you for this hint. In this particular case, we indeed (inadvertently) used the wrong wind speed limits to create the wind rose. We also re-checked the code for the calculation of the WFSS and could confirm the correct limits are implemented here. We adjusted the lower-middle panel of Fig. 2b and the corresponding text in the manuscript (now p11 L1-5).*

28) 9.16: 1th??
*Answer: We changed from 1th of August 2012 to 1st of August 2012 (now p11 L9).*

29) 9.24-29: This needs more interpretation here. What aspect of the dynamical model is failing? Is it the solution itself or is it the synoptic setup? Why does 8/1/2012 mostly succeed but this day fail?
*Answer: Also in this case, we mainly attribute the uniform wind directions simulated with the CCLM to the too strongly smoothed terrain in the model. For such events under low synoptic forcing, both wind fields show a too low spatial variability in wind direction. Regarding wind speed, the INCA wind field shows some variability with higher wind speeds in parts of the summit regions compared to wind speeds at lower altitude. Furthermore, a valley wind in the Enns valley becomes obvious. Probably the analysis part of the INCA model leads to a somewhat better representation of the wind field. We added additional text to draw attention to such effects (now p11 L18-26).*

*Could you please indicate what you mean with "Why does 8/1/2012 mostly succeed but this day fail"? We checked through the text but were not sure what's meant. For this CCLMvsINCA_therm_JBT event we are analyzing the 31th of Mai 2008 from 13:00-14:00.*

30) 10.6: Won't this always be true of COSMO in these synoptic circumstances? However, the scale of the features for the high wind regions here are actually above the coarser grid scales of COSMO, so this lack of resolution reasoning is not correct is it?
*Answer: For the WegenerNet FBR the wind fields are systematically overestimated which become obvious in the CCLMvsWN_strong_FBR case and in the statistical evaluation results (cf. also Fig. 4). For the WegenerNet JBT the low wind speeds are probably explained by negative orographic speed-up effects (Hewer, 1998) caused by a too smoothed terrain, compared to the WegnerNet FBR, where speed-up effects are leading to stronger wind speeds. For a detailed information about this speed-up effects see also point 2) in the responses to Major comments above. We added additional text about such effects to the 4.2*

*Statistical evaluation results section (for the indication of pages and lines of this text see also point 33 below, which deals with a similar question).*

31) 10.21: Unable instead of able??
*Answer: Yes, ok, done (now p12 L18).*

32) 10.25: Again, though isn't this the simpler terrain region?
*Answer: Yes, here we describe the performance of the CCLM in comparison to the INCA model for strong wind speeds for the hilly WegenerNet FBR. The influence of the terrain (e.g. channeling of air flow through the valleys) on the synoptic flow field is smaller in this hilly region than in the WegenerNet JBT region. That's why the CCLM shows similar performance as the INCA model despite the lower resolution for this region.*

33) Fig 4: It is very surprising that the COSMO model has a widespread systematic bias over the simpler FBR region, but a much reduced systematic bias in general over the much more complex terrain of the JBT region.
*Answer: Thank you for this hint; the difference in these bias values between the two regions is probably again attributed to the speed up effects. For more information please see point 2) in the response to Major comments above. Furthermore, it has to be noted that in comparison to the WegenerNet FBR region the INCA data and not the WegenerNet data were used as reference for the evaluation of the CCLM, due to missing WegenerNet data. Since the CCLM wind fields show small bias values for thermally induced wind events, compared to the INCA wind fields, similar results as in the CCLMvsINCA_therm_JBT case can be expected for a comparison of CCLM with WegenerNet data. In case of strong wind events, the intercomparison of the CCLM with the INCA model shows opposite patterns than the INCAvsWN_strong_JBT case, but with smaller bias values. Therefore the same bias values in attenuated form are to be expected for a comparison of the CCLM with WegenerNet data. We added this information to the text (now p14 L8-9, p14 L18-20).*

**Further changes in the manuscript**

*1) We now use the abbreviation CCLM instead of COSMO-CLM in the text and in all figures.*

*2) We have separated section "2.3 INCA and COSMO-CLM" data into "2.3 INCA data and 2.4 CCLM data".*

*3) Changed "138°E-17°E" to "13.8°E-17°E" (now p6 L10).*

*4) Corrected "at a defined station locations" to "at defined station locations" (now p8 L7).*

*5) Corrected "indicates fair weather conditions" to "indicate fair weather conditions" (now p8 L15).*

*6) Corrected "2N + 1" to "2N – 1" (now p9 L16).*

*7) Corrected "underpining" to "underpinning" (now p16 L7).*

*8) Corrected "expect" to "except" (now p16 L3).*

*9) Added additional text to the Acknowledgements section (now p17 L7-14).*

*10) Figure 3, (a) and (b): Changed "INCA resolution" to "INCA grid resolution" and "COSMO resolution" to "CCLM grid resolution".*

*11) Table 2: Corrected number of events for CCLMvsWN_therm_FBR from 1632 to 264.*

[revised manuscript text omitted]